# A Theoretical Framework for Deep and Locally Connected ReLU Network

## Abstract

Understanding theoretical properties of deep and locally connected nonlinear network, such as deep convolutional neural network (DCNN), is still a hard problem despite its empirical success. In this paper, we propose a novel theoretical framework for such networks with ReLU nonlinearity. The framework bridges data distribution with gradient descent rules, favors disentangled representations and is compatible with common regularization techniques such as Batch Norm, after a novel discovery of its projection nature. The framework is built upon teacher-student setting, by projecting the student's forward/backward pass onto the teacher's computational graph. We do not impose unrealistic assumptions (e.g., Gaussian inputs, independence of activation, etc). Our framework could help facilitate theoretical analysis of many practical issues, e.g. disentangled representations in deep networks.

## 1 Introduction

Deep Convolutional Neural Network (DCNN) has achieved a huge empirical success in multiple disciplines (e.g., computer vision (Krizhevsky et al., 2012; Simonyan & Zisserman, 2014; He et al., 2016), Computer Go (Silver et al., 2016; 2017; Tian & Zhu, 2016), and so on). On the other hand, its theoretical properties remain an open problem and an active research topic.

Learning deep models are often treated as non-convex optimization in a high-dimensional space. From this perspective, many properties in deep models have been analyzed: landscapes of loss functions (Choromanska et al., 2015b; Li et al., 2017; Mei et al., 2016), saddle points (Du et al., 2017; Dauphin et al., 2014), relationships between local minima and global minimum (Kawaguchi, 2016; Hardt & Ma, 2017; Safran & Shamir, 2017), trajectories of gradient descent (Goodfellow et al., 2014), path between local minima (Venturi et al., 2018), etc.

However, such a modeling misses two components: neither specific network structures nor input data distribution is considered. Both are critical in practice. Empirically, deep models work particular well for certain forms of data (e.g., images); theoretically, for certain data distribution, popular methods like gradient descent is shown to fail to recover network parameters (Brutzkus & Globerson, 2017).

Along this direction, previous theoretical works assume specific data distributions like spherical Gaussian and focus on shallow nonlinear networks (Tian, 2017; Brutzkus & Globerson, 2017; Du et al., 2018). These assumptions yield nice gradient forms and enable analysis of many properties such as global convergence. However, it is also nontrivial to extend such approaches to deep nonlinear neural networks that yield strong empirical performance.

In this paper, we propose a novel theoretical framework for deep and locally connected ReLU network that is applicable to general data distributions. Specifically, we embrace a teacher-student setting. The *teacher* computes classification labels via a computational graph that has local structures (e.g., CNN): intermediate variables in the graph, (called *summarization variables*), are computed from a subset of the input dimensions. The *student* network, with similar local structures, updates the weights to fit teacher's labels with gradient descent, without knowing the summarization variables.

One ultimate goal is to show that after training, each node in the student network is highly selective with respect to the summarization variable in the teacher. Achieving this goal will shed light to how the training of practically effective methods like CNN works, which remains a grand challenge. As a first step, we *reformulate* the forward/backward pass in gradient descent by marginalizing out the input data conditioned on the graph variables of the teacher at each layer. The reformulation has nice properties: **(1)** it relates data distribution with gradient update rules, **(2)** it is compatible with existing

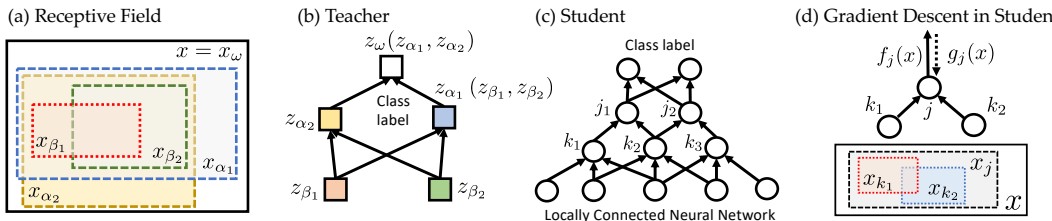

Figure 1: Problem setting. **(a)** We use Greek letters $\{\alpha, \beta, \ldots \omega\}$ to represent receptive fields. Receptive fields form a hierarchy. The entire input is denoted as $x$ (or $x_\omega$). A local region of an input $x$ is denoted as $x_\alpha$. **(b)** For each region $\alpha$, we have a latent multinomial discrete variable $z_\alpha$ which is computed from its immediate children $\{z_\beta\}_{\beta \in \mathrm{ch}(\alpha)}$. Given the input $x$, $z_\alpha = z_\alpha(x_\alpha)$ is a function of the image content $x_\alpha$ at $\alpha$. Finally, $z_\omega$ at the top level is the *class label*. **(c)** A locally connected neural network is trained with pairs $(x, z_\omega(x))$, where $z_\omega(x)$ is the class label generated from the teacher. **(d)** For each node $j$, $f_j(x)$ is the activation while $g_j(x)$ is the back-propagated gradient, both as function of input $x$ (and weights at different layers).

state-of-the-art regularization techniques such as Batch Normalization (Ioffe & Szegedy, 2015), and **(3)** it favors disentangled representation when data distributions have factorizable structures. To our best knowledge, our work is the first theoretical framework to achieve these properties for deep and locally connected nonlinear networks.

Previous works have also proposed framework to explain deep networks, e.g., renormalization group for restricted Boltzmann machines (Mehta & Schwab, 2014), spin-glass models (Amit et al., 1985; Choromanska et al., 2015a), transient chaos models (Poole et al., 2016), differential equations (Su et al., 2014; Saxe et al., 2013), information bottleneck (Achille & Soatto, 2017; Tishby & Zaslavsky, 2015; Saxe et al., 2018), etc. In comparison, our framework **(1)** imposes mild assumptions rather than unrealistic ones (e.g., independence of activations), **(2)** explicitly deals with back-propagation which is the dominant approach used for training in practice, and relates it with data distribution, and **(3)** considers spatial locality of neurons, an important component in practical deep models.

## 2 PROBLEM SETTING

We consider multi-layer (deep) and locally connected network with ReLU nonlinearity. We consider supervised setting, in which we have a dataset $\{(x, y)\}$, where $x$ is the input image and $y$ is its label computed from $x$ deterministically. It is hard to analyze $y$ which does not have a structure (e.g., random labels). Here our analysis assumes the generation of $y$ from $x$ has a specific hierarchical structure. We use teacher-student setting to study the property: a student network learns teacher's label $y$ via gradient descent, without knowing teacher's internal representations.

### 2.1 RECEPTIVE FIELDS

An interesting characteristics in locally connected network is that each neuron only covers a fraction of the input dimension. Furthermore, for deep and locally connected network, neurons in the lower layer cover a small region while neurons in the upper layer cover a large region.

We use Greek letters $\{\alpha, \beta, \ldots, \omega\}$ to represent receptive fields. For a receptive field $\alpha$, $x_\alpha$ is the content in that region. We use $\omega$ to represent the entire image (Fig. 1(a)).

Receptive fields form a hierarchy: $\alpha$ is a parent of $\beta$, denoted as $\alpha \in \mathrm{pa}(\beta)$ or $\beta \in \mathrm{ch}(\alpha)$, if $\alpha \supseteq \beta$ and there exists no other receptive field $\gamma \notin \{\alpha, \beta\}$ so that $\alpha \supseteq \gamma \supseteq \beta$. Note that siblings can have substantial overlaps (e.g., $\beta_1$ and $\beta_2$ in Fig. 1(a)). With this partial ordering, we can attach layer number $l$ to each receptive field: $\alpha \in \mathrm{pa}(\beta)$ implies $l(\beta) = l(\alpha) + 1$. For top-most layer (closest to classification label), $l = 0$ and for bottom-most layer, $l = L$.

For locally connected network, a neuron (or node) $j \in \alpha$ means its receptive field is $\alpha$. Denote $n_\alpha$ as the number of nodes covering the same region (e.g., multi-channel case, Fig. 2(a)). The image content is $x_{\alpha(j)}$, abbreviated as $x_j$ if no ambiguity. The parent $j$'s receptive field covers its children's.

### 2.2 THE TEACHER

We assume the label $y$ of the input $x$ is computed by a *teacher* in a bottom-up manner: for each region $\alpha$, we compute a *summarization* variable $z_\alpha$ from the summarization variables of its children:

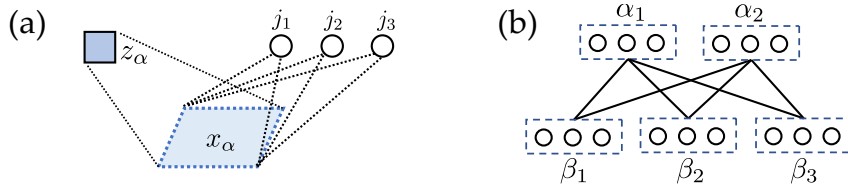

Figure 2: **(a)** Multiple nodes (neurons) share the same receptive field $\alpha$. Note that $n_\alpha$ is the number of nodes sharing the receptive field $\alpha$. **(b)** Grouping nodes with the same receptive fields together. By abuse of notation, $\alpha$ also represents the collection of all nodes with the same receptive field.

$z_\alpha = \phi_\alpha(\{z_\beta\}_{\beta \in \mathrm{ch}(\alpha)})$. This procedure is repeated until the top-level summarization $z_\omega$ is computed, which is the class label $y$. We denote $\phi = \{\phi_\alpha\}$ as the collection of all summarization functions.

For convenience, we assume $z_\alpha$ be discrete variables that takes $m_\alpha$ possible values. Intuitively, $m_\alpha$ is exponential w.r.t the area of the receptive field $\mathrm{sz}(\alpha)$, for binary input, $m_\alpha \leq 2^{\mathrm{sz}(\alpha)}$. We call a particular assignment of $z_\alpha$, $z_\alpha = a$, an *event*. For the bottom-most layers, $z$ is just the (discretized) value in each dimension.

At each stage, the upward function is deterministic but *lossy*: $z_\alpha$ does not contain all the information in $\{z_\beta\}$ for $\beta \in \mathrm{ch}(\alpha)$. Indeed, it keeps relevant information in the input region $x_\alpha$ with respect to the class label, and discards the irrelevant part. During training, all summarization variables $\mathcal{Z} = \{z_\alpha\}$ are unknown to the student, except for the label $y$.

**Example of teacher networks**. Locally connected network itself is a good example of teacher network, in which nodes of different channels located at one specific spatial location form some encoding of the variable $z_\alpha$. Note that the relationship between a particular input $x$ and the corresponding values of the summarization variable $z$ at each layer is purely deterministic.

The reason why probabilistic quantities (e.g., $\mathbb{P}(z_\alpha)$ and $\mathbb{P}(z_\alpha|z_\beta)$) appear in our formulation, is due to marginalization over $z$ (or $x$). This marginalization implicitly establishes a relationship between the conditional probabilities $\mathbb{P}(z_\alpha|z_\beta)$ and the input data distribution $\mathbb{P}(x)$. If we have specified $\mathbb{P}(z_\alpha|z_\beta)$ at each layer, then we implicitly specify a certain kind of data distribution $\mathbb{P}(x)$. Conversely, given a certain kind of $\mathbb{P}(x)$ and summarization function $\phi$, we can compute $\mathbb{P}(z_\alpha|z_\beta)$ by sampling $x$, compute summarization variable $z_\alpha$, and accumulate frequency statistics of $\mathbb{P}(z_\alpha|z_\beta)$. If there is an overlap between sibling receptive fields, then it is likely that some relationship among $\mathbb{P}(z_\alpha|z_\beta)$ might exist, which we leave for future work.

Although such an indirect specification may not be as intuitive and mathematically easy to deal with as common assumptions used in previous works (e.g., assuming Gaussian input (Tian, 2017; Du et al., 2018; Brutzkus & Globerson, 2017)), it gives much more flexibility of the distribution $x$ and is more likely to be true empirically.

**Comparison with top-down generative model**. An alternative (and more traditional) way to specify data distribution is to use a top-down generative model: first sample the label $y$, then sample the latent variables $z_\alpha$ at each layer in a top-down manner, until the input layer. Marginalizing over all the latent variables $z_\alpha$ yields a class-conditioned data distribution $\mathbb{P}(x|y)$.

The main difficulty of this top-down modeling is that when the receptive fields $\alpha$ and $\alpha'$ of sibling latent variables overlap, the underlying graphical model becomes loopy. This makes the population loss function, which involves an integral over the input data $x$, very difficult to deal with. As a result, it is nontrivial to find a concise relationship between the parameters in the top-down modeling (e.g., conditional probability) and the optimization techniques applied to neural network (e.g., gradient descent). In contrast, as we will see in Sec. 3, our modeling naturally gives relationship between gradient descent rules and conditional probability between nearby summarization variables.

### 2.3 THE STUDENT

We consider a neuron (or node) $j$. Denote $f_j$ as its activation after nonlinearity and $g_j$ as the (input) gradient it receives after filtered by ReLU's gating (Fig. 1(d)). Note that both $f_j$ and $g_j$ are deterministic functions of the input $x$ and label $y$, and are abbreviated as $f_j(x)$ and $g_j(x)$. [1].

---

[1]Note that all analysis still holds with bias terms. We omit them for brevity.

The activation $f_j$ and gradient $g_k$ can be written as (note that $f_j'$ is the binary gating function):

$$f_j(x) = f_j'(x) \sum_{k \in \text{ch}(j)} w_{jk} f_k(x), \quad g_k(x) = f_k'(x) \sum_{j \in \text{pa}(k)} w_{jk} g_j(x) \tag{1}$$

And the weight update for gradient descent is $\Delta w_{jk} = \mathbb{E}_x \left[ f_k(x) g_j(x) \right]$. Here is the expectation is with respect to a training dataset (or a batch), depending on whether GD or SGD has been used. We also use $f_j^{\text{raw}}$ and $g_j^{\text{raw}}$ as the counterpart of $f_j$ and $g_j$ before nonlinearity.

For locally connected network, the activation $f_j$ of node $j$ is only dependent on the region $x_j$, rather than the entire image $x$. This means that $f_j(x) = f_j(x_j)$ and $f_j(x_j) = f_j'(x_j) \sum_k w_{jk} f_k(x_k)$. However, the gradient $g_j$ is determined by the entire image $x$, and its label $y$, i.e., $g_j = g_j(x, y)$. Note that since the label $y$ is a deterministic (but unknown) function of $x$, for gradient we just write $g_j = g_j(x)$.

**Marginalized Gradient.** For locally connected network, the gradient $g_j$ has some nice structures. From Eqn. 17 we knows that $\Delta w_{jk} = \mathbb{E}_x \left[ f_k(x) g_j(x) \right] = \mathbb{E}_{x_k} \left[ f_k(x_k) \mathbb{E}_{x_{-k}|x_k} \left[ g_j(x) \right] \right]$. Define $x_{-k} = x \backslash x_k$ as the input image $x$ except for $x_k$. Then we can define the *marginalized gradient*:

$$g_j(x_k) = \mathbb{E}_{x_{-k}|x_k} \left[ g_j(x) \right] \tag{2}$$

as the marginalization (average) of $x_{-k}$, while keep $x_k$ fixed. With this notation, we can write $\Delta w_{jk} = \mathbb{E}_{x_k} \left[ f_k(x_k) g_j(x_k) \right]$.

On the other hand, the gradient which back-propagates to a node $k$ can be written as

$$g_k(x) = f_k'(x) \sum_{j \in \text{pa}(k)} w_{jk} g_j(x) = f_k'(x_k) \sum_j w_{jk} g_j(x) \tag{3}$$

where $f_k'$ is the derivative of activation function of node $k$ (for ReLU it is just a gating function). If we take expectation with respect to $x_{-k}|x_k$ on both side, we get

$$g_k(x_k) = f_k'(x_k) g_k^{\text{raw}}(x_k) = f_k'(x_k) \sum_{j \in \text{pa}(k)} w_{jk} g_j(x_k) \tag{4}$$

Note that all marginalized gradients $g_j(x_k)$ are independently computed by marginalizing with respect to all regions that are outside the receptive field $x_k$. Interestingly, there is a relationship between these gradients that respects the locality structure:

**Theorem 1** (Recursive Property of marginalized gradient). $g_j(x_k) = \mathbb{E}_{x_{j,-k}|x_k} \left[ g_j(x_j) \right]$

This shows that there is a recursive structure in marginal gradient: we can first compute $g_j(x_j)$ for top node $j$, then by marginalizing over the region within $x_j$ but outside $x_k$, we get its projection $g_j(x_k)$ on child $k$, then by Eqn. 20 we collect all projections from all the parents of node $k$, to get $g_k(x_k)$. This procedure can be repeated until we arrive at the leaf nodes.

## 3 THE FRAMEWORK AND ITS GOAL

Let's first consider the following quantity. For each neural node $j$, we want to compute the expected gradient given a particular *factor* $z_\alpha$, where $\alpha = \text{rf}(j)$ (the reception field of node $j$):

$$g_j(z_\alpha) \equiv \mathbb{E}_{X_j|z_\alpha} \left[ g_j(X_j) \right] = \int g_j(x_j) \mathbb{P}(x_j|z_\alpha) \mathrm{d}x_j \tag{5}$$

And $\tilde{g}_j(z_\alpha) = g_j(z_\alpha) \mathbb{P}(z_\alpha)$. Similarly, $f_j(z_\alpha) = \mathbb{E}_{X_j|z_\alpha} \left[ f_j(X_j) \right]$ and $f_j'(z_\alpha) = \mathbb{E}_{X_j|z_\alpha} \left[ f_j'(X_j) \right]$.

Note that $\mathbb{P}(x_j|z_\alpha)$ is the *frequency count* of $x_j$ for $z_\alpha$. If $z_\alpha$ captures all information of $x_j$, then $\mathbb{P}(x_j|z_\alpha)$ is a delta function. Throughout the paper, we use frequentist interpretation of probabilities.

**Goal.** Intuitively, if we have $g_j(z_\alpha = a) > 0$ and $g_j(z_\alpha \neq a) < 0$, then the node $j$ learns about the *hidden* event $z_\alpha = a$. For multi-class classification, the top level nodes (just below the softmax layer) already embrace such correlations (here $j$ is the class label): $g_j(y = j) > 0$ and $g_j(y \neq j) < 0$, where we know $z_\omega = y$ is the top level factor. A natural question now arises:

*Does gradient descent automatically push $g_j(z_\alpha)$ to be correlated with the factor $z_\alpha$?*

| | Dimension | Description |
|---|---|---|
| $F_\alpha, \tilde{G}_\alpha, D_\alpha$ | $m_\alpha$-by-$n_\alpha$ | Activation $f_j(z_\alpha)$, gradient $\tilde{g}_j(z_\alpha)$ and gating prob $f'_j(z_\alpha)$ at group $\alpha$. |
| $W_{\beta\alpha}$ | $n_\beta$-by-$n_\alpha$ | Weight matrix that links group $\alpha$ and $\beta$ |
| $P_{\alpha\beta}$ | $m_\alpha$-by-$m_\beta$ | Prob $\mathbb{P}(z_\beta\|z_\alpha)$ of events at group $\alpha$ and $\beta$ |

Table 1: Matrix Notation. See Eqn. 8 and Eqn. 59.

If this is true, then gradient descent on deep models is essentially a *weak-supervised* approach that automatically learns the intermediate events at different levels. Giving a complete answer of this question is very difficult and is beyond the scope of this paper. *As a first step*, we build a theoretical framework that enables such analysis. We start with the relationship between neighboring layers:

**Theorem 2** (Reformulation). *For node $j$ and $k$ and their receptive field $\alpha$ and $\beta$. If the following two conditions holds:*

1) **Onsite Conditional independence**. $\mathbb{P}(x_j|z_{\mathrm{rf}(j)}, z_{...}) = \mathbb{P}(x_j|z_{\mathrm{rf}(j)})$.

2) **Decorrelation**. *Given $z_\beta$, $g_k^{\mathrm{raw}}(\cdot)$ and $f_k^{\mathrm{raw}}(\cdot)$ are* uncorrelated *with $f'_k(\cdot)$:*
$$\mathbb{E}_{x_k|z_\beta}\left[f'_k g_k^{\mathrm{raw}}\right] = \mathbb{E}_{x_k|z_\beta}\left[f'_k\right]\mathbb{E}_{x_k|z_\beta}\left[g_k^{\mathrm{raw}}\right], \quad \mathbb{E}_{x_k|z_\beta}\left[f'_k f_k^{\mathrm{raw}}\right] = \mathbb{E}_{x_k|z_\beta}\left[f'_k\right]\mathbb{E}_{x_k|z_\beta}\left[f_k^{\mathrm{raw}}\right] \quad (6)$$

*Then the following iterative equations hold:*
$$f_j(z_\alpha) = f'_j(z_\alpha)\sum_{k\in\mathrm{ch}(j)} w_{jk}\mathbb{E}_{z_\beta|z_\alpha}\left[f_k(z_\beta)\right], \quad g_k(z_\beta) = f'_k(z_\beta)\sum_{j\in\mathrm{pa}(k)} w_{jk}\mathbb{E}_{z_\alpha|z_\beta}\left[g_j(z_\alpha)\right] \quad (7)$$

The reformulation becomes *exact* if $z_\alpha$ contains all information of the region.

**Theorem 3.** *If $\mathbb{P}(x_j|z_\alpha)$ is a delta function for all $\alpha$, then all conditions in Thm. 2 hold.*

While Thm. 3 holds in the ideal (and maybe trivial) case, both assumptions are still practically reasonable. For assumption (1), the main idea is that the image content $x_\alpha$ is most related to the summarization variable $z_\alpha$ located at the same receptive field $\alpha$, and less related to others. On the other hand, assumptions (2) holds approximately if the summarization variable is fine-grained.

Intuitively, $\mathbb{P}(x_j|z_\alpha)$ is a distribution encoding how much information gets *lost* if we only know the factor $z_\alpha$. Climbing up the ladder, more and more information is lost while keeping the critical part for the classification. This is consistent with empirical observations (Bau et al., 2017), in which the low-level features in DCNN are generic, and high-level features are more class-specific.

One key property of this formulation is that, it relates conditional probabilities $\mathbb{P}(z_\alpha, z_\beta)$, and thus input data distribution $\mathbb{P}(x)$ into the gradient descent rules. This is important since running backpropagation on different dataset is now formulated into the same framework with different probability, i.e., frequency counts of events. By studying which family of distribution leads to the desired property, we could understand backpropagation better.

Furthermore, the property of stochastic gradient descent (SGD) can be modeled as using an imperfect estimate $\tilde{\mathbb{P}}(z_\alpha, z_\beta)$ of the true probability $\mathbb{P}(z_\alpha, z_\beta)$ when running backpropagation. This is because each batch is a rough sample of the data distribution so the resulting $\mathbb{P}(z_\alpha, z_\beta)$ will also be different. This could also unite GD and SGD analysis.

For boundary conditions, in the lowest level $L$, we could treat each input pixel (or a group of pixels) as a single event: $f_k(z_\beta) = \mathbb{I}[k = z_\beta]$. For top level, each node $j$ corresponds to a class label $j$ while the summarization variable $z_\alpha$ also take class labels: $g_j(z_\alpha) = a_1\mathbb{I}[j = z_\alpha] - a_2\mathbb{I}[j \neq z_\alpha]$.

If we group the nodes with the same reception field at the same level together (Fig. 2), we have the matrix form of Eqn. 7 ($\circ$ is element-wise multiplication):

**Theorem 4** (Matrix Representation of Reformulation).
$$F_\alpha = D_\alpha \circ \sum_{\beta\in\mathrm{ch}(\alpha)} P_{\alpha\beta}F_\beta W_{\beta\alpha}, \quad \tilde{G}_\beta = D_\beta \circ \sum_{\alpha\in\mathrm{pa}(\beta)} P_{\alpha\beta}^T\tilde{G}_\alpha W_{\beta\alpha}^T, \quad \Delta W_{\beta\alpha} = (P_{\alpha\beta}F_\beta)^T\tilde{G}_\alpha \quad (8)$$

See Tbl. 3 for the notation. For this dynamics, we want $F_\omega^* = I_{n_\omega}$, i.e., the top $n_\omega$ neurons faithfully represents the classification labels. Therefore, the top level gradient is $G_\omega = I_{n_\omega} - F_\omega$. On the other side, for each region $\beta$ at the bottom layer, we have $F_\beta = I_{n_\beta}$, i.e., the input contains all the preliminary factors. For all regions $\alpha$ in the top-most and bottom-most layers, we have $n_\alpha = m_\alpha$.

(a) Network before/after adding BN     (b) Forward/backward in BN     (c) Projected Gradient

Figure 3: Batch Normalization (BN) as a projection. **(a)** Add BN by inserting a new node $j_{\text{bn}}$. **(b)** Forward/backward pass in BN and relevant quantities. **(c)** The gradient $\mathbf{g}$ that is propagated down is a projection of input gradient $\mathbf{g}_{\text{bn}}$ onto the orthogonal complementary space spanned by $\{\mathbf{f}, \mathbf{1}\}$.

## 4    BATCH NORMALIZATION UNDER REFORMULATION

Our reformulation naturally incorporates empirical regularization technique like Batch Normalization (BN) (Ioffe & Szegedy, 2015).

### 4.1    BATCH NORMALIZATION AS A PROJECTION

We start with a novel finding of Batch Norm: the back-propagated gradient through Batch Norm layer at a node $j$ is a projection onto the orthogonal complementary subspace spanned by all one vectors and the current activations of node $j$.

Denote pre-batchnorm activations as $\mathbf{f} = [f_j(x_1), \dots f_j(x_N)]$ where $N$ is the batchsize. In Batch Norm, $\mathbf{f}$ is whitened to be $\tilde{\mathbf{f}}$, then linearly transformed to yield the output $\mathbf{f}_{\text{bn}}$ (note that we omit node subscript $j$ for clarity):

$$\hat{\mathbf{f}} = \mathbf{f} - \mu\mathbf{1}, \quad \tilde{\mathbf{f}} = \hat{\mathbf{f}}/\sigma, \quad \mathbf{f}_{\text{bn}} = c_1\tilde{\mathbf{f}} + c_0 \tag{9}$$

where $\mu = \frac{1}{N}\mathbf{f}^T\mathbf{1}$ and $\sigma^2 = \frac{1}{N}\hat{\mathbf{f}}^T\hat{\mathbf{f}}$ and $c_1, c_0$ are learnable parameters.

The original Batch Norm paper (Ioffe & Szegedy, 2015) derives complicated and unintuitive weight update rules. With vector notation, the update has a compact form with a clear geometric meaning.

**Theorem 5** (Backpropagation of Batch Norm). *For a top-down pre-BN gradient $\mathbf{g}_{\text{bn}}$ (a vector of size $N$-by-1,$N$ is the batchsize), the gradient after passing BN layer is the following:*

$$\mathbf{g} = J^{BN}(\mathbf{f})\mathbf{g}_{\text{bn}} = \frac{c_1}{\sigma}P_{\mathbf{f},\mathbf{1}}^{\perp}\mathbf{g}_{\text{bn}}, \quad \mathbf{g_c} \equiv [g_{c_1}, g_{c_0}]^T = S(\mathbf{f})^T\mathbf{g}_{\text{bn}} \tag{10}$$

*Here $P_{\mathbf{f},\mathbf{1}}^{\perp}$ is the orthogonal complementary projection onto subspace $\{\mathbf{f}, \mathbf{1}\}$ and $S(\mathbf{f}) \equiv [\tilde{\mathbf{f}}, \mathbf{1}]$.*

Intuitively, the back-propagated gradient $\mathbf{g}$ is zero-mean and perpendicular to the input activation $\mathbf{f}$ of BN layer, as illustrated in Fig. 3. Unlike (Kohler et al., 2018) that analyzes BN in an approximate manner, in Thm. 5 we do not impose any assumptions.

### 4.2    BATCH NORM UNDER THE REFORMULATION

In our reformulation, we take the expectation of input $x$ so there is no explicit notation of batch. However, we could regard each sample in the batch as i.i.d. samples from the data distribution $\mathbb{P}(x)$. Then the analysis of Batch Norm in Sec. 4.1 could be applied in the reformulation and yield similar results, using the quantity that $\mathbb{E}_x[f_j(x)] = \mathbb{E}_{z_\alpha}[f_j(z_\alpha)]$.

In this case, we have $\mu = \mathbb{E}_{z_\alpha}[f_j]$ and $\sigma^2 = \mathbb{E}_{z_\alpha}[(f_j(z_\alpha) - \mu)^2]$, and $J^{BN}(\mathbf{f}) = \frac{c_1}{\sigma}P_{\mathbf{f},\mathbf{1}}^{\perp,z_\alpha}$. Note that the projection matrix $P_{\mathbf{f},\mathbf{1}}^{\perp,z_\alpha}$ is under the new inner product $\langle f_j, g_j \rangle_{z_\alpha} = \mathbb{E}_{z_\alpha}[f_j(z_\alpha)g_j(z_\alpha)]$ and norm $\|f\|_{z_\alpha} = \langle f, f \rangle_{z_\alpha}^{1/2}$. In comparison, Sec. 4.1 is a special case with $\mathbb{P}(z_\alpha) = \frac{1}{N}\sum_{i=1}^N \delta(z_\alpha - \phi_\alpha(x_i))$, where $x_1, \dots, x_N$ are the batch samples.

One consequence is that for $\tilde{G}_\alpha$, we have $\mathbf{1}^T\tilde{G}_\alpha = [\mathbb{E}_{z_\alpha}[g_1(z_\alpha)], \dots, \mathbb{E}_{z_\alpha}[g_{n_\alpha}(z_\alpha)]] = \mathbf{0}$ since $g_j(\cdot)$ is in the null space of $\mathbf{1}$ under the inner product $\langle \cdot, \cdot \rangle_{z_\alpha}$. This property will be used in Sec. 5.2.

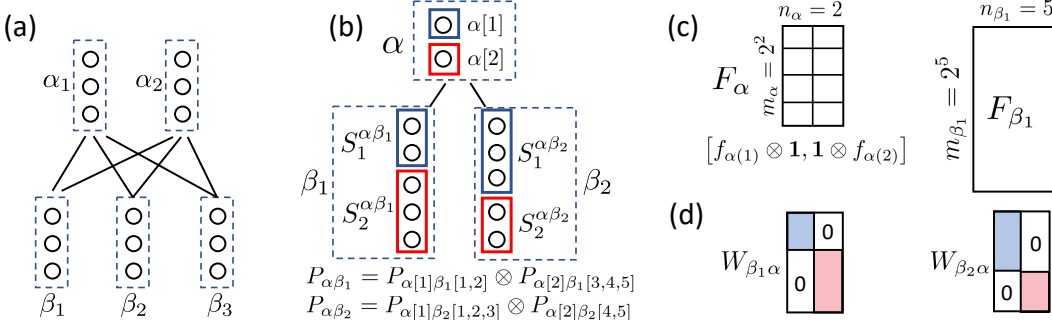

Figure 4: Disentangled representation. **(a)** Nodes are grouped according to regions. **(b)** An example of one parent region $\alpha$ (2 nodes) and two child regions $\beta_1$ and $\beta_2$ (5 nodes each). We assume factorization property of data distribution $P$. **(c)** disentangled activations, **(d)** Separable weights.

## 5 EXAMPLE APPLICATIONS OF PROPOSED THEORETICAL FRAMEWORK

With the help of the theoretical framework, we now can analyze interesting structures of gradient descent in deep models, when the data distribution $\mathbb{P}(z_\alpha, z_\beta)$ satisfies specific conditions. Here we give two concrete examples: the role played by nonlinearity and in which condition disentangled representation can be achieved. Besides, from the theoretical framework, we also give general comments on multiple issues (e.g., overfitting, GD versus SGD) in deep learning.

### 5.1 NONLINEAR VERSUS LINEAR

In the formulation, $m_\alpha$ is the number of possible events within a region $\alpha$, which is often exponential with respect to the size $\mathrm{sz}(\alpha)$ of the region. The following analysis shows that a linear model cannot handle it, even with exponential number of nodes $n_\alpha$, while a nonlinear one with ReLU can.

**Definition 1** (Convex Hull of a Set). *We define the convex hull* $\mathrm{Conv}(P)$ *of $m$ points $P \subset \mathbb{R}^n$ to be* $\mathrm{Conv}(P) = \left\{ P\mathbf{a}, \mathbf{a} \in \mathbf{\Delta}^{n-1} \right\}$, *where* $\mathbf{\Delta}^{n-1} = \{\mathbf{a} \in \mathbb{R}^n, a_i \geq 0, \sum_i a_i = 1\}$. *A row $p_j$ is called vertex if $p_j \notin \mathrm{Conv}(P \backslash p_j)$.*

**Definition 2.** *A matrix $P$ of size $m$-by-$n$ is called $k$-vert, or $\mathrm{vert}(P) = k \leq m$, if its $k$ rows are vertices of the convex hull generated by its rows. $P$ is called all-vert if $k = m$.*

**Theorem 6** (Expressibility of ReLU Nonlinearity). *Assuming $m_\alpha = n_\alpha = \mathcal{O}(\exp(\mathrm{sz}(\alpha)))$, where $\mathrm{sz}(\alpha)$ is the size of receptive field of $\alpha$. If each $P_{\alpha\beta}$ is all-vert, then: ($\omega$ is top-level receptive field)*

$$\min_W Loss_{\mathrm{ReLU}}(W) = 0, \quad \min_W Loss_{\mathrm{Linear}}(W) = \mathcal{O}(\exp(\mathrm{sz}(\omega))) \tag{11}$$

Note that here $Loss(W) \equiv \|F_\omega - I\|_F^2$. This shows the power of nonlinearity, which guarantees full rank of output, even if the matrices involved in the multiplication are low-rank. The following theorem shows that for intermediate layers whose input is not identity, the all-vert property remains.

**Theorem 7.** **(1)** *If $F$ is full row rank, then $\mathrm{vert}(PF) = \mathrm{vert}(P)$.* **(2)** *$PF$ is all-vert iff $P$ is all-vert.*

This means that if all $P_{\alpha\beta}$ are all-vert and its input $F_\beta$ is full-rank, then with the same construction of Thm. 6, $F_\alpha$ can be made identity. In particular, if we sample $W$ randomly, then with probability 1, all $F_\beta$ are full-rank, in particular the top-level input $F_1$. Therefore, using top-level $W_1$ alone would be sufficient to yield zero generalization error, as shown in the previous works that random projection could work well.

### 5.2 DISENTANGLED REPRESENTATION

The analysis in Sec. 5.1 assumes that $n_\alpha = m_\alpha$, which means that we have sufficient nodes, *one neuron for one event*, to convey the information forward to the classification level. In practice, this is never the case. When $n_\alpha \ll m_\alpha = \mathcal{O}(\exp(\mathrm{sz}(\alpha)))$ and the network needs to represent the information in a proper way so that it can be sent to the top level. Ideally, if the factor $z_\alpha$ can be written down as a list of binary factors: $z_\alpha = \left[ z_{\alpha[1]}, z_{\alpha[2]}, \ldots, z_{\alpha[j]} \right]$, the output of a node $j$ could represent $z_{\alpha[j]}$, so that all $m_\alpha$ events can be represented concisely with $n_\alpha$ nodes.

| Symbols | Description |
|---------|-------------|
| $z_{\alpha[j]}$ | The $j$-th binary factor of region $\alpha$. $z_{\alpha[j]}$ can take 0 or 1. |
| $\mathbf{p}_{\alpha[j]}$ | 2-by-1 marginal probability vector of binary factor $z_{\alpha[j]}$. |
| $\mathbf{f}_{\alpha[j]}, \mathbf{g}_{\alpha[j]}, \tilde{\mathbf{g}}_{\alpha[j]}$ | The $j$-th column of $F_\alpha$, $G_\alpha$ and $\tilde{G}_\alpha$ corresponding to $j$-th binary factor $z_{\alpha[j]}$. |
| $\mathbf{1}$ / $\mathbf{0}$ | All-1 / All-0 vector. Its dimension depends on context. |
| $F_1 \otimes F_2$ | Out (or tensor) product of $F_1$ and $F_2$. $\otimes_{k=1}^K F_k$ is the product of multiple $F_k$. |
| $\{S_i^{\alpha\beta}\}$ | Disjoint collection of $n_\alpha$ indices sets. $S_i^{\alpha\beta}$ are the indices of downstream nodes in $\beta$ to $i$-th binary factor in $\alpha$ (Fig. 4(b)). |
| $W_{\beta\alpha}[S_j^{\alpha\beta}, j]$ | The $j$-th subcolumn of weight matrix $W_{\beta\alpha}$, whose rows are selected by $S_j^{\alpha\beta}$. |

Table 2: Symbols used in Sec. 5.2.

To come up with a complete theory for disentangled representation in deep nonlinear network is far from trivial and beyond the scope of this paper. In the following, we make an initial attempt by constructing factorizable $P_{\alpha\beta}$ so that disentangled representation is possible in the forward pass. First we need to formally define what is disentangled representation:

**Definition 3.** *The activation $F_\alpha$ is* disentangled, *if its $j$-th column $F_{\alpha,:j} = \mathbf{1} \otimes \ldots \otimes \mathbf{f}_{\alpha[j]} \otimes \ldots \otimes \mathbf{1}$, where each $\mathbf{f}_{\alpha[j]}$ and $\mathbf{1}$ is a 2-by-1 vector.*

**Definition 4.** *The gradient $\tilde{G}_\alpha$ is* disentangled, *if its $j$-th column $\tilde{G}_{\alpha,:j} = \mathbf{p}_{\alpha[1]} \otimes \ldots \otimes \tilde{\mathbf{g}}_{\alpha[j]} \otimes \ldots \otimes \mathbf{p}_{\alpha[n_\alpha]}$, where $\mathbf{p}_{\alpha[j]} = [\mathbb{P}(\alpha[j] = 0), \mathbb{P}(\alpha[j] = 1)]^T$ and $\tilde{\mathbf{g}}_{\alpha[j]}$ is a 2-by-1 vector.*

Intuitively, this means that each node $j$ represents the binary factor $z_\alpha[j]$. A follow-up question is whether such disentangled properties carries over layers in the forward pass. It turns out that the disentangled structure carries if the data distribution and weights have compatible structures:

**Definition 5.** *The weights $W_{\beta\alpha}$ is* separable *with respect to a disjoint set $\{S_i^{\alpha\beta}\}$, if $W_{\beta\alpha} = \mathrm{diag}\left(W_{\beta\alpha}[S_1^{\alpha\beta}, 1], W_{\beta\alpha}[S_2^{\alpha\beta}, 2], \ldots, W_{\beta\alpha}[S_{n_\alpha}^{\alpha\beta}, n_\alpha]\right)$.*

**Theorem 8** (Disentangled Forward). *If for each $\beta \in \mathrm{ch}(\alpha)$, $P_{\alpha\beta}$ can be written as a tensor product $P_{\alpha\beta} = \bigotimes_i P_{\alpha[i]\beta[S_i^{\alpha\beta}]}$ where $\{S_i^{\alpha\beta}\}$ are $\alpha\beta$-dependent disjointed set, $W_{\beta\alpha}$ is separable with respect to $\{S_i^{\alpha\beta}\}$, $F_\beta$ is disentangled, then $F_\alpha$ is also disentangled (with/without ReLU /Batch Norm).*

If the bottom activations are disentangled, by induction, all activations will be disentangled. The next question is whether gradient descent preserves such a structure. The answer is also conditionally yes:

**Theorem 9** (Separable Weight Update). *If $P_{\alpha\beta} = \bigotimes_i P_{\alpha[i]\beta[S_i]}$, $F_\beta$ and $\tilde{G}_\alpha$ are both disentangled, $\mathbf{1}^T \tilde{G}_\alpha = \mathbf{0}$, then the gradient update $\Delta W_{\beta\alpha}$ is separable with respect to $\{S_i\}$.*

Therefore, with disentangled $F_\beta$ and $\tilde{G}_\alpha$ and centered gradient $\mathbf{1}^T \tilde{G}_\alpha = \mathbf{0}$, the separable structure is conserved over gradient descent, given the initial $W_{\beta\alpha}^{(0)}$ is separable. Note that centered gradient is guaranteed if we insert Batch Norm (Eqn. 83) after linear layers. And the activation $F$ remains disentangled if the weights are separable.

The hard part is whether $\tilde{G}_\beta$ remains disentangled during backpropagation, if $\{\tilde{G}_\alpha\}_{\alpha \in \mathrm{pa}(\beta)}$ are all disentangled. If so, then the disentangled representation is self-sustainable under gradient descent. This is a non-trivial problem and generally requires structures of data distribution. We put some discussion in the Appendix and leave this topic for future work.

## 6 Explanation of common behaviors in Deep Learning

In the proposed formulation, the input $x$ in Eqn. 7 is integrated out, and the data distribution is now encoded into the probabilistic distribution $\mathbb{P}(z_\alpha, z_\beta)$, and their marginals. A change of such distribution means the input distribution has changed. For the first time, we can now analyze many practical factors and behaviors in the DL training that is traditionally not included in the formulation.

**Over-fitting.** Given finite number of training samples, there is always error in estimated factor-factor distribution $\tilde{\mathbb{P}}(z_\alpha, z_\beta)$ and factor-observation distribution $\tilde{\mathbb{P}}(x_\alpha | z_\alpha)$. In some cases, a slight change of distribution would drastically change the optimal weights for prediction, which is overfitting.

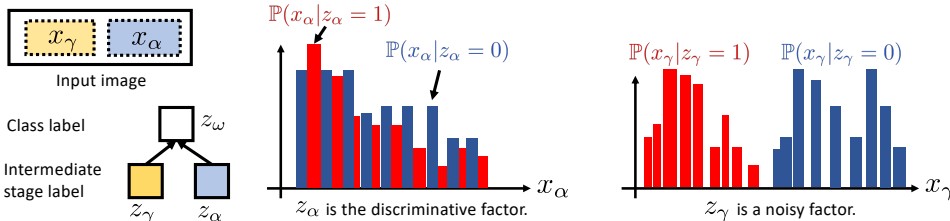

Figure 5: Overfitting Example

Here is one example. Suppose there are two different kinds of events at two disjoint reception fields: $z_\alpha$ and $z_\gamma$. The class label is $z_\omega$, which equals $z_\alpha$ but is not related to $z_\gamma$. Therefore, we have:

$$\tilde{\mathbb{P}}(z_\omega = 1 | z_\alpha = 1) = 1, \quad \tilde{\mathbb{P}}(z_\omega = 1 | z_\alpha = 0) = 0 \tag{12}$$

Although $z_\gamma$ is unrelated to the class label $z_\omega$, with finite samples $z_\gamma$ could show spurious correlation:

$$\tilde{\mathbb{P}}(z_\omega = 1 | z_\gamma = 1) = 0.5 + \epsilon, \quad \tilde{\mathbb{P}}(z_\omega = 1 | z_\gamma = 0) = 0.5 - \epsilon \tag{13}$$

On the other hand, as shown in Fig. 5, $\mathbb{P}(x_\alpha | z_\alpha)$ contains a lot of detailed structures and is almost impossible to separate in the finite sample case, while $\mathbb{P}(x_\gamma | z_\gamma)$ could be well separated for $z_\gamma = 0/1$. Therefore, for node $j$ with $\mathrm{rf}(j) = \alpha$, $f_j(z_\alpha) \approx \text{constant}$ (input almost indistinguishable):

$$\Delta w_j = \mathbb{E}_{z_\alpha} [f_j(z_\alpha) g_0(z_\alpha)] \approx 0 \tag{14}$$

where $g_0(z_\alpha) = \mathbb{E}_{z_\omega | z_\alpha} [g_0(z_\omega)] = \begin{cases} 1 & z_\alpha = 1 \\ -1 & z_\alpha = 0 \end{cases}$ , which is a strong gradient signal backpropagated from the top softmax level, since $z_\alpha$ is strongly correlated with $z_\omega$. For node $k$ with $\mathrm{rf}(k) = \gamma$, an easy separation of the input (e.g., random initialization) yields distinctive $f_k(z_\gamma)$. Therefore,

$$\Delta w_k = \mathbb{E}_{z_\gamma} [f_j(z_\gamma) g_0(z_\gamma)] > 0 \tag{15}$$

where $g_0(z_\gamma) = \mathbb{E}_{z_\omega | z_\gamma} [g_0(z_\omega)] = \begin{cases} 2\epsilon & z_\gamma = 1 \\ -2\epsilon & z_\gamma = 0 \end{cases}$ , a weak signal because of $z_\gamma$ is (almost) unrelated to the label. Therefore, we see that the weight $w_j$ that links to meaningful receptive field $z_\alpha$ does not receive strong gradient, while the weight $w_k$ that links to irrelevant (but spurious) receptive field $z_\gamma$ receives strong gradient. This will lead to overfitting.

With more data, over-fitting is alleviated since (1) $\tilde{\mathbb{P}}(z_\omega | z_\gamma)$ becomes more accurate and $\epsilon \to 0$; (2) $\tilde{\mathbb{P}}(x_\alpha | z_\alpha)$ starts to show statistical difference for $z_\alpha = 0/1$ and thus $f_j(z_\alpha)$ shows distinctiveness.

Note that there exists a **second** explanation: we could argue that $z_\gamma$ is a *true* but *weak* factor that contributes to the label, while $z_\alpha$ is a *fictitious* discriminative factor, since the appearance difference between $z_\alpha = 0$ and $z_\alpha = 1$ (i.e., $\tilde{\mathbb{P}}(x_\alpha | z_\alpha)$ for $\alpha = 0/1$) could be purely due to noise and thus should be neglected. With finite number of samples, these two cases are essentially indistinguishable. Models with different induction bias might prefer one to the other, yielding drastically different generalization error. For neural network, SGD prefers the second explanation but if under the pressure of training, it may also explore the first one by pushing gradient down to distinguish subtle difference in the input. This may explain why the same neural networks can fit random-labeled data, and generalize well for real data (Zhang et al., 2016).

**Gradient Descent: Stochastic or not?** Previous works (Keskar et al., 2017) show that empirically stochastic gradient decent (SGD) with small batch size tends to converge to "flat" minima and offers better generalizable solution than those uses larger batches to compute the gradient.

From our framework, SGD update with small batch size is equivalent to using a perturbed/noisy version of $\mathbb{P}(z_\alpha, z_\beta)$ at each iteration. Such an approach naturally reduces aforementioned over-fitting issues, which is due to hyper-sensitivity of data distribution and makes the final weight solution invariant to changes in $\mathbb{P}(z_\alpha, z_\beta)$, yielding a "flat" solution.

# 7 CONCLUSION AND FUTURE WORK

In this paper, we propose a novel theoretical framework for deep (multi-layered) nonlinear network with ReLU activation and local receptive fields. The framework utilizes the specific structure of

neural networks, and formulates input data distributions explicitly. Compared to modeling deep models as non-convex problems, our framework reveals more structures of the network; compared to recent works that also take data distribution into considerations, our theoretical framework can model deep networks without imposing idealistic analytic distribution of data like Gaussian inputs or independent activations. Besides, we also analyze regularization techniques like Batch Norm, depicts its underlying geometrical intuition, and shows that BN is compatible with our framework.

Using this novel framework, we have made an initial attempt to analyze many important and practical issues in deep models, and provides a novel perspective on overfitting, generalization, disentangled representation, etc. We emphasize that in this work, we barely touch the surface of these core issues in deep learning. As a future work, we aim to explore them in a deeper and more thorough manner, by using the powerful theoretical framework proposed in this paper.

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

## 8 APPENDIX

### 8.1 PROBLEM SETTING

We consider a neuron (or node) $j$. Denote $f_j$ as its activation after nonlinearity and $g_j$ as the (input) gradient it receives after filtered by ReLU's gating. Note that both $f_j$ and $g_j$ are deterministic functions of the input $x$ and label $y$. Since $y$ is a deterministic function of $x$, we can write $f_j = f_j(x)$ and $g_j = g_j(x)$. Note that all analysis still holds with bias terms. We omit them for brevity.

The activation $f_j$ and gradient $g_k$ can be written as (note that $f'_j$ is the binary gating function):

$$f_j(x) = f'_j(x) \sum_{k \in \mathrm{ch}(j)} w_{jk} f_k(x), \quad g_k(x) = f'_k(x) \sum_{j \in \mathrm{pa}(k)} w_{jk} g_j(x) \tag{16}$$

And the weight update for gradient descent is:

$$\Delta w_{jk} = \mathbb{E}_x \left[ f_k(x) g_j(x) \right] \tag{17}$$

Here is the expectation is with respect to a training dataset (or a batch), depending on whether GD or SGD has been used. We also use $f_j^{\mathrm{raw}}$ and $g_j^{\mathrm{raw}}$ as the counterpart of $f_j$ and $g_j$ before nonlinearity.

### 8.2 MARGINALIZED GRADIENT

Given the structure of locally connected network, the gradient $g_j$ has some nice structures. From Eqn. 17 we knows that $\Delta w_{jk} = \mathbb{E}_x \left[ f_k(x) g_j(x) \right] = \mathbb{E}_{x_k} \left[ f_k(x_k) \mathbb{E}_{x_{-k}|x_k} \left[ g_j(x) \right] \right]$. Define $x_{-k} = x \backslash x_k$ as the input image $x$ except for $x_k$. Then we can define the *marginalized gradient*:

$$g_j(x_k) = \mathbb{E}_{x_{-k}|x_k} \left[ g_j(x) \right] \tag{18}$$

as the marginalization (average) of $x_{-k}$, while keep $x_k$ fixed. With this notation, we can write $\Delta w_{jk} = \mathbb{E}_{x_k} \left[ f_k(x_k) g_j(x_k) \right]$.

On the other hand, the gradient which back-propagates to a node $k$ can be written as

$$g_k(x) = f'_k(x) \sum_{j \in \mathrm{pa}(k)} w_{jk} g_j(x) = f'_k(x_k) \sum_j w_{jk} g_j(x) \tag{19}$$

where $f'_k$ is the derivative of activation function of node $k$ (for ReLU it is just a gating function). If we take expectation with respect to $x_{-k}|x_k$ on both side, we get

$$g_k(x_k) = f'_k(x_k) g_k^{\mathrm{raw}}(x_k) = f'_k(x_k) \sum_{j \in \mathrm{pa}(k)} w_{jk} g_j(x_k) \tag{20}$$

Note that all marginalized gradients $g_j(x_k)$ are independently computed by marginalizing with respect to all regions that are outside the receptive field $x_k$. Interestingly, there is a relationship between these gradients that respects the locality structure:

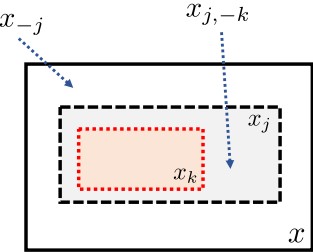

Figure 6: Notation used in Thm. 1.

**Theorem 1** (Recursive Property of marginalized gradient).

$$g_j(x_k) = \mathbb{E}_{x_{j,-k}|x_k} \left[ g_j(x_j) \right] \tag{21}$$

*Proof.* We have:

$$
\begin{aligned}
g_j(x_k) &= \mathbb{E}_{x_{-k}|x_k}\left[g_j(x)\right] \\
&= \mathbb{E}_{x_{-j},x_{j,-k}|x_k}\left[g_j(x)\right] \\
&= \mathbb{E}_{x_{-j}|x_{j,-k},x_k}\left[\mathbb{E}_{x_{j,-k}|x_k}\left[g_j(x)\right]\right] \\
&= \mathbb{E}_{x_{-j}|x_j}\left[\mathbb{E}_{x_{j,-k}|x_k}\left[g_j(x)\right]\right] \\
&= \mathbb{E}_{x_{j,-k}|x_k}\left[\mathbb{E}_{x_{-j}|x_j}\left[g_j(x)\right]\right] \\
&= \mathbb{E}_{x_{j,-k}|x_k}\left[g_j(x_j)\right]
\end{aligned}
$$

$\square$

## 8.3 Network theorem

**Theorem 2** (Reformulation). *Denote $\alpha = \mathrm{rf}(j)$ and $\beta = \mathrm{rf}(k)$. $k$ is a child of $j$. If the following two conditions hold:*

- ***Focus of knowledge.*** $\mathbb{P}(x_k|z_\alpha, z_\beta) = \mathbb{P}(x_k|z_\beta)$.

- ***Broadness of knowledge.*** $\mathbb{P}(x_j|z_\alpha, z_\beta) = \mathbb{P}(x_j|z_\alpha)$.

- ***Decorrelation.*** *Given $z_\beta$, $(g_k^{\mathrm{raw}}(\cdot)$ and $f_k'(\cdot))$ and $(f_k^{\mathrm{raw}}(\cdot)$ and $f_k'(\cdot))$ are* uncorrelated

*Then the following two conditions holds:*

$$
f_j(z_\alpha) = f_j'(z_\alpha) \sum_{k\in\mathrm{ch}(j)} w_{jk}\mathbb{E}_{z_\beta|z_\alpha}\left[f_k(z_\beta)\right] \tag{22a}
$$

$$
g_k(z_\beta) = f_k'(z_\beta) \sum_{j\in\mathrm{pa}(k)} w_{jk}\mathbb{E}_{z_\alpha|z_\beta}\left[g_j(z_\alpha)\right] \tag{22b}
$$

*Proof.* For Eqn. 22a, we have:

$$
f_j^{\mathrm{raw}}(z_\alpha) = \int f_j^{\mathrm{raw}}(x)\mathbb{P}(x|z_\alpha)\mathrm{d}x \tag{23}
$$

$$
= \int f_j^{\mathrm{raw}}(x_j)\mathbb{P}(x_j|z_\alpha)\mathrm{d}x_j \tag{24}
$$

$$
= \int \left(\sum_{k\in\mathrm{ch}(j)} w_{jk}f_k(x_k)\right)\mathbb{P}(x_j|z_\alpha)\mathrm{d}x_j \tag{25}
$$

And for each of the entry, we have:

$$
\int f_k(x_k)\mathbb{P}(x_j|z_\alpha)\mathrm{d}x_j = \int f_k(x_k)\mathbb{P}(x_k|z_\alpha)\mathrm{d}x_k \tag{26}
$$

For $\mathbb{P}(x_k|z_\alpha)$, using focus of knowledge, we have:

$$
\mathbb{P}(x_k|z_\alpha) = \sum_{z_\beta} \mathbb{P}(x_k, z_\beta|z_\alpha) \tag{27}
$$

$$
= \sum_{z_\beta} \mathbb{P}(x_k|z_\beta, z_\alpha)\mathbb{P}(z_\beta|z_\alpha) \tag{28}
$$

$$
= \sum_{z_\beta} \mathbb{P}(x_k|z_\beta)\mathbb{P}(z_\beta|z_\alpha) \tag{29}
$$

Therefore, following Eqn. 26, we have:

$$\int f_k(x_k)\mathbb{P}(x_k|z_\alpha)\mathrm{d}x_k = \int f_k(x_k)\sum_{z_\beta}\mathbb{P}(x_k|z_\beta)\mathbb{P}(z_\beta|z_\alpha)\mathrm{d}x_k \tag{30}$$

$$= \sum_{z_\beta}\left(\int f_k(x_k)\mathbb{P}(x_k|z_\beta)\mathrm{d}x_k\right)\mathbb{P}(z_\beta|z_\alpha) \tag{31}$$

$$= \sum_{z_\beta} f_k(z_\beta)\mathbb{P}(z_\beta|z_\alpha) \tag{32}$$

$$= \mathbb{E}_{z_\beta|z_\alpha}[f_k(z_\beta)] \tag{33}$$

Putting it back to Eqn. 25 and we have:

$$f_j^{\mathrm{raw}}(z_\alpha) = \sum_{k\in\mathrm{ch}(j)} w_{jk}\mathbb{E}_{z_\beta|z_\alpha}[f_k(z_\beta)] \tag{34}$$

For Eqn. 22b, similarly we have:

$$g_k^{\mathrm{raw}}(z_\beta) = \int g_k^{\mathrm{raw}}(x)\mathbb{P}(x|z_\beta)\mathrm{d}x \tag{35}$$

$$= \int \sum_{j\in\mathrm{pa}(k)} w_{jk}g_j(x)\mathbb{P}(x|z_\beta)\mathrm{d}x \tag{36}$$

Notice that we have:

$$\mathbb{P}(x|z_\beta) = \mathbb{P}(x_j|z_\beta)\mathbb{P}(x_{-j}|x_j, z_\beta) = \mathbb{P}(x_j|z_\beta)\mathbb{P}(x_{-j}|x_j) \tag{37}$$

since $x_j$ covers $x_k$ which determines $z_\beta$. Therefore, for each item we have:

$$\int g_j(x)\mathbb{P}(x|z_\beta)\mathrm{d}x = \int g_j(x)\mathbb{P}(x_j|z_\beta)\mathbb{P}(x_{-j}|x_j)\mathrm{d}x \tag{38}$$

$$= \int \left(\int g_j(x)\mathbb{P}(x_{-j}|x_j)\mathrm{d}x_{-j}\right)\mathbb{P}(x_j|z_\beta)\mathrm{d}x_j \tag{39}$$

$$= \int g_j(x_j)\mathbb{P}(x_j|z_\beta)\mathrm{d}x_j \tag{40}$$

Then we use the broadness of knowledge:

$$\mathbb{P}(x_j|z_\beta) = \sum_{z_\alpha}\mathbb{P}(x_j, z_\alpha|z_\beta) \tag{41}$$

$$= \sum_{z_\alpha}\mathbb{P}(x_j|z_\alpha, z_\beta)\mathbb{P}(z_\alpha|z_\beta) \tag{42}$$

$$= \sum_{z_\alpha}\mathbb{P}(x_j|z_\alpha)\mathbb{P}(z_\alpha|z_\beta) \tag{43}$$

Following Eqn. 40, we now have:

$$\int g_j(x)\mathbb{P}(x|z_\beta)\mathrm{d}x = \int g_j(x_j)\sum_{z_\alpha}\mathbb{P}(x_j|z_\alpha)\mathbb{P}(z_\alpha|z_\beta)\mathrm{d}x_j \tag{44}$$

$$= \sum_{z_\alpha}\left(\int g_j(x_j)\mathbb{P}(x_j|z_\alpha)\mathrm{d}x_j\right)\mathbb{P}(z_\alpha|z_\beta) \tag{45}$$

$$= \sum_{z_\alpha} g_j(z_\alpha)\mathbb{P}(z_\alpha|z_\beta) \tag{46}$$

$$= \mathbb{E}_{z_\alpha|z_\beta}[g_j(z_\alpha)] \tag{47}$$

Putting it back to Eqn. 36 and we have:

$$g_k^{\text{raw}}(z_\beta) = \sum_{j \in \text{pa}(k)} w_{jk} \mathbb{E}_{z_\alpha | z_\beta} [g_j(z_\alpha)] \tag{48}$$

Using the definition of $g_k(z_\beta)$:

$$g_k(z_\beta) = \int g_k(x_k) \mathbb{P}(x_k | z_\beta) \mathrm{d}x_k \tag{49}$$

$$= \int f_k'(x_k) g_k^{\text{raw}}(x_k) \mathbb{P}(x_k | z_\beta) \mathrm{d}x_k \tag{50}$$

$$= \mathbb{E}_{X_k | z_\beta} [f_k'(X_k) g_k^{\text{raw}}(X_k)] \tag{51}$$

The un-correlation between $g_k^{\text{raw}}(\cdot)$ and $f_k'(\cdot)$ means that

$$\mathbb{E}_{X_k | z_\beta} [f_k' g_k^{\text{raw}}] = \mathbb{E}_{X_k | z_\beta} [f_k'] \cdot \mathbb{E}_{X_k | z_\beta} [g_k^{\text{raw}}] \tag{52}$$

Similarly for $f_j(z_\alpha)$. $\qquad\square$

## 8.4 EXACTNESS OF REFORMULATION

The following theorem shows that the reformulation is exact if $z_\alpha$ has all information of the region.

**Theorem 3.** *If $\mathbb{P}(x_j | z_\alpha)$ is a delta function for all $\alpha$, then the conditions of Thm. 2 hold and the reformulation becomes exact.*

*Proof.* The fact that $\mathbb{P}(x_j | z_\alpha)$ is a delta function means that there exists a function $\phi_j$ so that:

$$\mathbb{P}(x_j | z_\alpha) = \delta(x_j - \phi_j(z_\alpha)) \tag{53}$$

That is, $z_\alpha$ contains all information of $x_j$ (or $x_\alpha$). Therefore,

- **Broadness of knowledge.** $z_\alpha$ contains strictly more information than $z_\beta$ for $\beta \in \text{ch}(\alpha)$, therefore $\mathbb{P}(x_j | z_\alpha, z_\beta) = \mathbb{P}(x_j | z_\alpha)$.

- **Focus of knowledge.** $z_\beta$ captures all information of $z_k$, so $\mathbb{P}(x_k | z_\alpha, z_\beta) = \mathbb{P}(x_k | z_\beta)$.

- **Decorrelation.** For any $h_1(x_j)$ and $h_2(x_j)$ we have

$$\mathbb{E}_{X_j | z_\alpha} [h_1 h_2] = \int h_1(x_j) h_2(x_j) \mathbb{P}(x_j | z_\alpha) \mathrm{d}x_j \tag{54}$$

$$= \int h_1(x_j) h_2(x_j) \delta(x_j - \phi_j(z_\alpha)) \mathrm{d}x_j \tag{55}$$

$$= h_1(\phi_j(z_\alpha)) h_2(\phi_j(z_\alpha)) \tag{56}$$

$$= \int h_1(x_j) \mathbb{P}(x_j | z_\alpha) \mathrm{d}x_j \int h_2(x_j) \mathbb{P}(x_j | z_\alpha) \mathrm{d}x_j \tag{57}$$

$$= \mathbb{E}_{X_j | z_\alpha} [h_1] \mathbb{E}_{X_j | z_\alpha} [h_2] \tag{58}$$

$\qquad\square$

## 8.5 MATRIX FORM

**Theorem 4** (Matrix Representation of Reformulation)**.**

$$F_\alpha = D_\alpha \circ \sum_{\beta \in \text{ch}(\alpha)} P_{\alpha\beta} F_\beta W_{\beta\alpha}, \quad \tilde{G}_\beta = D_\beta \circ \sum_{\alpha \in \text{pa}(\beta)} P_{\alpha\beta}^T \tilde{G}_\alpha W_{\beta\alpha}^T, \quad \Delta W_{\beta\alpha} = (P_{\alpha\beta} F_\beta)^T \tilde{G}_\alpha \tag{59}$$

| | Dimension | Description |
|---|---|---|
| $F_\alpha, D_\alpha$ | $m_\alpha$-by-$n_\alpha$ | Activation $f_j(z_\alpha)$ and gating prob $f_j'(z_\alpha)$ in group $\alpha$. |
| $G_\alpha, \tilde{G}_\alpha$ | $m_\alpha$-by-$n_\alpha$ | Gradient $g_j(z_\alpha)$ and unnormalized gradient $\tilde{g}_j(z_\alpha)$ in group $\alpha$. |
| $W_{\beta\alpha}$ | $n_\beta$-by-$n_\alpha$ | Weight matrix that links group $\beta$ and $\alpha$. |
| $P_{\alpha\beta}, P_{\alpha\beta}^b$ | $m_\alpha$-by-$m_\beta$ | Prob $\mathbb{P}(z_\beta\|z_\alpha)$, $\mathbb{P}(z_\alpha\|z_\beta)$ of events between group $\beta$ and $\alpha$. |
| $\Lambda_\alpha$ | $m_\alpha$-by-$m_\alpha$ | Diagonal matrix encoding prior prob $\mathbb{P}(z_\alpha)$. |

Table 3: Matrix Notation. See Eqn. 59.

*Proof.* We first consider one certain group $\alpha$ and $\beta$, which uses $x_\alpha$ and $x_\beta$ as the receptive field. For this pair, we can write Eqn. 22 in the following matrix form:

$$F_\alpha^{\text{raw}} = P_{\alpha\beta} F_\beta W_{\beta\alpha} \tag{60a}$$

$$F_\alpha = F_\alpha^{\text{raw}} \circ D_\alpha \tag{60b}$$

$$G_\beta^{\text{raw}} = \left(P_{\alpha\beta}^b\right)^T G_\beta W_{\beta\alpha}^T \tag{60c}$$

$$G_\beta = G_\beta^{\text{raw}} \circ D_\beta \tag{60d}$$

Using $\Lambda_\beta (P_{\alpha\beta}^b)^T = P_{\alpha\beta}^T \Lambda_\alpha$ and $\tilde{G}_\alpha = \Lambda_\alpha G_\alpha$, we could simplify Eqn. 60 as follows:

$$F_\alpha = P_{\alpha\beta} F_\beta W_{\beta\alpha} \circ D_\alpha \tag{61a}$$

$$\tilde{G}_\beta = (P_{\alpha\beta})^T \tilde{G}_\beta W_{\beta\alpha}^T \circ D_\beta \tag{61b}$$

Therefore, using the fact that $\sum_{j\in\text{pa}(k)} = \sum_{\alpha\in\text{pa}(\beta)} \sum_{j\in\alpha}$ (where $\beta = \text{rf}(k)$) and $\sum_{k\in\text{ch}(j)} = \sum_{\beta\in\text{ch}(\alpha)} \sum_{k\in\beta}$ (where $\alpha = \text{rf}(j)$), and group all nodes that share the receptive field together, we have:

$$F_\alpha = D_\alpha \circ \sum_{\beta\in\text{ch}(\alpha)} P_{\alpha\beta} F_\beta W_{\beta\alpha} \tag{62a}$$

$$\tilde{G}_\beta = D_\beta \circ \sum_{\alpha\in\text{pa}(\beta)} P_{\alpha\beta}^T \tilde{G}_\alpha W_{\beta\alpha}^T \tag{62b}$$

For the gradient update rule, from Eqn. 17 notice that:

$$\Delta w_{jk} = \mathbb{E}_x\left[f_k(x)g_j(x)\right] \tag{63}$$

$$= \int f_k(x)g_j(x)\mathbb{P}(x)\mathrm{d}x \tag{64}$$

$$= \int f_k(x)g_j(x)\sum_{z_\alpha}\mathbb{P}(x|z_\alpha)\mathbb{P}(z_\alpha)\mathrm{d}x \tag{65}$$

$$= \sum_{z_\alpha}\int f_k(x)g_j(x)\mathbb{P}(x|z_\alpha)\mathbb{P}(z_\alpha)\mathrm{d}x \tag{66}$$

We assume decorrelation so we have:

$$\Delta w_{jk} = \sum_{z_\alpha}\mathbb{E}_{X|z_\alpha}\left[f_k(x)\right]g_j(z_\alpha)\mathbb{P}(z_\alpha) \tag{67}$$

$$= \sum_{z_\alpha}\mathbb{E}_{X_k|z_\alpha}\left[f_k(x_k)\right]\tilde{g}_j(z_\alpha) \tag{68}$$

For $\mathbb{E}_{X_k|z_\alpha}[f_k(x_k)]$, again we use focus of knowledge:

$$\mathbb{E}_{X_k|z_\alpha}[f_k(x_k)] = \int f_k(x_k)\mathbb{P}(x_k|z_\alpha)\mathrm{d}x_k \tag{69}$$

$$= \sum_{z_\beta} \int f_k(x_k)\mathbb{P}(x_k|z_\alpha, z_\beta)\mathbb{P}(z_\beta|z_\alpha)\mathrm{d}x_k \tag{70}$$

$$= \sum_{z_\beta} \int f_k(x_k)\mathbb{P}(x_k|z_\beta)\mathbb{P}(z_\beta|z_\alpha)\mathrm{d}x_k \tag{71}$$

$$= \sum_{z_\beta} f_k(z_\beta)\mathbb{P}(z_\beta|z_\alpha) \tag{72}$$

Put them together and we have:

$$\Delta w_{jk} = \sum_{z_\alpha}\sum_{z_\beta} f_k(z_\beta)\tilde{g}_j(z_\alpha)\mathbb{P}(z_\beta|z_\alpha) = \mathbb{E}_{z_\alpha,z_\beta}[f_k(z_\beta)g_j(z_\alpha)] \tag{73}$$

Write it in concise matrix form and we get:

$$\Delta W_{\beta\alpha} = (P_{\alpha\beta}F_\beta)^T \tilde{G}_\alpha \tag{74}$$

$\square$

## 8.6 BATCH NORM AS A PROJECTION

**Theorem 5** (Backpropagation of Batch Norm). *For a top-down gradient* $\mathbf{g}$*, BN layer gives the following gradient update (*$P_{\mathbf{f},\mathbf{1}}^\perp$ *is the orthogonal complementary projection of subspace* $\{\mathbf{f},\mathbf{1}\}$*):*

$$\mathbf{g} = J^{BN}(\mathbf{f})\mathbf{g}_{\mathrm{bn}} = \frac{c_1}{\sigma}P_{\mathbf{f},\mathbf{1}}^\perp\mathbf{g}_{\mathrm{bn}}, \quad \mathbf{g_c} = S(\mathbf{f})^T\mathbf{g}_{\mathrm{bn}} \tag{75}$$

*Proof.* We denote pre-batchnorm activations as $f^{(i)} = f_j(x_i)$ $(i = 1\dots N)$. In Batch Norm, $f^{(i)}$ is whitened to be $\tilde{f}^{(i)}$, then linearly transformed to yield the output $f_{\mathrm{bn}}^{(i)}$:

$$\hat{f}^{(i)} = f^{(i)} - \mu, \quad \tilde{f}^{(i)} = \hat{f}^{(i)}/\sigma, \quad f_{\mathrm{bn}}^{(i)} = c_1\tilde{f}^{(i)} + c_0 \tag{76}$$

where $\mu = \frac{1}{N}\sum_i f^{(i)}$ and $\sigma^2 = \frac{1}{N}\sum_i(f^{(i)} - \mu)^2$ and $c_1, c_0$ are learnable parameters.

While in the original batch norm paper, the weight update rules are super complicated and unintuitive (listed here for a reference):

$$\frac{\partial\ell}{\partial\hat{x}_i} = \frac{\partial\ell}{\partial y_i} \cdot \gamma$$

$$\frac{\partial\ell}{\partial\sigma_{\mathcal{B}}^2} = \sum_{i=1}^m \frac{\partial\ell}{\partial\hat{x}_i} \cdot (x_i - \mu_{\mathcal{B}}) \cdot \frac{-1}{2}(\sigma_{\mathcal{B}}^2 + \epsilon)^{-3/2}$$

$$\frac{\partial\ell}{\partial\mu_{\mathcal{B}}} = \left(\sum_{i=1}^m \frac{\partial\ell}{\partial\hat{x}_i} \cdot \frac{-1}{\sqrt{\sigma_{\mathcal{B}}^2+\epsilon}}\right) + \frac{\partial\ell}{\partial\sigma_{\mathcal{B}}^2} \cdot \frac{\sum_{i=1}^m -2(x_i-\mu_{\mathcal{B}})}{m}$$

$$\frac{\partial\ell}{\partial x_i} = \frac{\partial\ell}{\partial\hat{x}_i} \cdot \frac{1}{\sqrt{\sigma_{\mathcal{B}}^2+\epsilon}} + \frac{\partial\ell}{\partial\sigma_{\mathcal{B}}^2} \cdot \frac{2(x_i-\mu_{\mathcal{B}})}{m} + \frac{\partial\ell}{\partial\mu_{\mathcal{B}}} \cdot \frac{1}{m}$$

$$\frac{\partial\ell}{\partial\gamma} = \sum_{i=1}^m \frac{\partial\ell}{\partial y_i} \cdot \hat{x}_i$$

$$\frac{\partial\ell}{\partial\beta} = \sum_{i=1}^m \frac{\partial\ell}{\partial y_i}$$

Figure 7: Original BN rule from (Ioffe & Szegedy, 2015).

It turns out that with vector notation, the update equations have a compact vector form with clear geometric meaning.

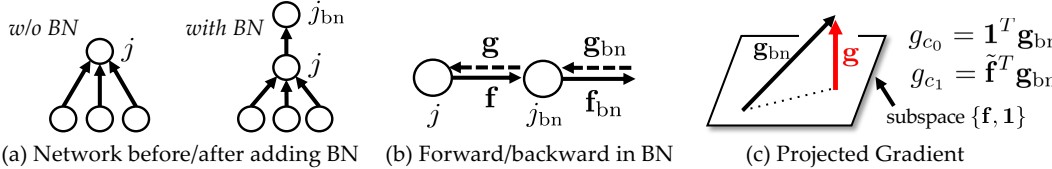

(a) Network before/after adding BN     (b) Forward/backward in BN     (c) Projected Gradient

Figure 8: Batch Normalization (BN) as a projection. **(a)** Add BN by inserting a new node $j_{\text{bn}}$. **(b)** Forward/backward pass in BN and relevant quantities. **(c)** The gradient $\mathbf{g}$ that is propagated down is a projection of input gradient $\mathbf{g}_{\text{bn}}$ onto the orthogonal complementary space spanned by $\{\mathbf{f}, \mathbf{1}\}$.

To achieve that, we first write down the vector form of forward pass of batch normalization:

$$\hat{\mathbf{f}} = P_1^\perp \mathbf{f}, \quad \tilde{\mathbf{f}} = \hat{\mathbf{f}}/\|\hat{\mathbf{f}}\|_{\text{uni}}, \quad \mathbf{f}_{\text{bn}} = c_1\tilde{\mathbf{f}} + c_0 \mathbf{1} = S(\mathbf{f})\mathbf{c} \tag{77}$$

where $\mathbf{f}$, $\hat{\mathbf{f}}$, $\tilde{\mathbf{f}}$ and $\mathbf{f}_{\text{bn}}$ are vectors of size $N$, $P_1^\perp \equiv I - \frac{\mathbf{1}\mathbf{1}^T}{N}$ is the projection matrix that centers the data, $\sigma = \|\mathbf{f}\|_{\text{uni}} = \frac{1}{\sqrt{N}}\|\mathbf{f}\|_2$ and $\mathbf{c} \equiv [c_1, c_0]^T$ are the parameters in Batch Normalization and $S(\mathbf{f}) \equiv [\tilde{\mathbf{f}}, \mathbf{1}]$ is the standardized data. Note that $S(\mathbf{f})^T S(\mathbf{f}) = N \cdot I_2$ ($I_2$ is 2-by-2 identity matrix) and thus $S(\mathbf{x})$ is an column-orthogonal $N$-by-2 matrix. If we put everything together, then we have:

$$\mathbf{f}_{\text{bn}} = c_1 \frac{P_1^\perp \mathbf{f}}{\|P_1^\perp \mathbf{f}\|_{\text{uni}}} + c_0 \mathbf{1} \tag{78}$$

Using this notation, we can compute the Jacobian of batch normalization layer. Specifically, for any vector $\mathbf{f}$, we have:

$$\frac{\mathrm{d}\left(\frac{\mathbf{f}}{\|\mathbf{f}\|}\right)}{\mathrm{d}\mathbf{f}} = \frac{1}{\|\mathbf{f}\|}\left(I - \frac{\mathbf{f}\mathbf{f}^T}{\|\mathbf{f}\|^2}\right) = \frac{1}{\|\mathbf{f}\|}P_{\mathbf{f}}^\perp \tag{79}$$

where $P_{\mathbf{f}}^\perp$ projects a vector into the *orthogonal complementary* space of $\mathbf{f}$. Therefore we have:

$$J^{BN}(\mathbf{f}) = \frac{\mathrm{d}\mathbf{f}_{\text{bn}}}{\mathrm{d}\mathbf{f}} = c_1\frac{\mathrm{d}\tilde{\mathbf{f}}}{\mathrm{d}\mathbf{f}} = c_1\frac{\mathrm{d}\hat{\mathbf{f}}}{\mathrm{d}\mathbf{f}}\frac{\mathrm{d}\tilde{\mathbf{f}}}{\mathrm{d}\hat{\mathbf{f}}} = \frac{c_1}{\sigma}P_{\tilde{\mathbf{f}},\mathbf{1}}^c \tag{80}$$

where $P_{\tilde{\mathbf{f}},\mathbf{1}}^\perp = I - \frac{S(\mathbf{f})S(\mathbf{f})^T}{N}$ is a symmetric projection matrix that projects the input gradient to the orthogonal complement space spanned by $\tilde{\mathbf{x}}$ and $\mathbf{1}$ (Fig. 3(b)). Note that the space spanned by $\tilde{\mathbf{f}}$ and $\mathbf{1}$ is also the space spanned by $\mathbf{f}$ and $\mathbf{1}$, since $\tilde{\mathbf{f}} = (\mathbf{f} - \mu\mathbf{1})/\sigma$ can be represented linearly by $\mathbf{f}$ and $\mathbf{1}$. Therefore $P_{\tilde{\mathbf{f}},\mathbf{1}}^\perp = P_{\mathbf{f},\mathbf{1}}^\perp$.

An interesting property is that since $\mathbf{f}_{\text{bn}}$ returns a vector in the subspace of $\mathbf{f}$ and $\mathbf{1}$, for the $N$-by-$N$ Jacobian matrix of Batch Normalization, we have:

$$J^{BN}(\mathbf{f})\mathbf{f}_{\text{bn}} = J^{BN}(\mathbf{f})\mathbf{1} = J^{BN}(\mathbf{f})\mathbf{f} = \mathbf{0} \tag{81}$$

Following the backpropagation rule, we get the following gradient update for batch normalization. If $\mathbf{g}_{\text{bn}} = \partial L/\partial \bar{\mathbf{f}}$ is the gradient from top, then

$$\mathbf{g_c} = S(\mathbf{f})^T \mathbf{g}_{\text{bn}}, \qquad \mathbf{g} = J^{BN}(\mathbf{f})\mathbf{g}_{\text{bn}} \tag{82}$$

Therefore, any gradient (vector of size $N$) that is back-propagated to the input of BN layer will be automatically orthogonal to that activation (which is also a vector of size $N$). $\qquad\square$

### 8.6.1 BATCH NORMALIZATION IN THE REFORMULATION

The analysis of Batch Norm is compatible with the reformulation and we arrive at similar backpropagation rule, by noticing that $\mathbb{E}_x[f_j(x)] = \mathbb{E}_{z_\alpha}[f_j(z_\alpha)]$:

$$\mu = \mathbb{E}_{z_\alpha}[f_j], \quad \sigma^2 = \mathbb{E}_{z_\alpha}\left[(f_j(z_\alpha) - \mu)^2\right], \quad J^{BN}(\mathbf{f}) = \frac{c_1}{\sigma}P_{\mathbf{f},\mathbf{1}}^\perp \tag{83}$$

Note that we still have the projection property, but under the new inner product $\langle f_j, g_j \rangle_{z_\alpha} = \mathbb{E}_{z_\alpha}[f_j(z_\alpha)g_j(z_\alpha)]$ and norm $\|f\|_{z_\alpha} = \langle f, f \rangle_{z_\alpha}^{1/2}$.

### 8.6.2 CONSERVED QUANTITY IN BATCH NORMALIZATION FOR RELU ACTIVATION

One can find an interesting quantity, by multiplying $g_j(x)$ on both side of the forward equation in Eqn. 16 and taking expectation:

$$\mathbb{E}_x\left[g_j f_j\right] = \mathbb{E}_x\left[\sum_{k\in\mathrm{ch}(j)} w_{jk} f_k g_j\right] = \sum_{k\in\mathrm{ch}(j)} w_{jk}\Delta w_{jk} \tag{84}$$

Using the language of differential equation, we know that:

$$\int_0^t \mathbb{E}_x\left[g_j^{(t')} f_j^{(t')}\right]\mathrm{d}t' = E_j(t) - E_j(0) \tag{85}$$

where $E_j = \frac{1}{2}\sum_{k\in\mathrm{ch}(j)} w_{jk}^2 = \frac{1}{2}\|W_{j\cdot}\|^2$. If we place Batch Normalization layer just after ReLU activation and linear layer, by BN property, since $\mathbb{E}_x\left[g_j f_j\right] \equiv 0$ for all iterations, the row energy $E_j(t)$ of weight matrix $W$ of the linear layer is conserved over time. This might be part of the reason why BN helps stabilize the training. Otherwise energy might "leak" from one layer to nearby layers.

### 8.7 EXAMPLE APPLICATIONS OF PROPOSED THEORETICAL FRAMEWORK

With the help of the theoretical framework, we now can analyze interesting structures of gradient descent in deep models, when the data distribution $\mathbb{P}(z_\alpha, z_\beta)$ satisfies specific conditions. Here we give two concrete examples: the role played by nonlinearity and in which condition disentangled representation can be achieved. Besides, from the theoretical framework, we also give general comments on multiple issues (e.g., overfitting, GD versus SGD) in deep learning.

#### 8.7.1 NONLINEAR VERSUS LINEAR

In the formulation, $m_\alpha$ is the number of possible events within a region $\alpha$, which is often exponential with respect to the size $\mathrm{sz}(\alpha)$ of the region. The following analysis shows that a linear model cannot handle it, even with exponential number of nodes $n_\alpha$, while a nonlinear one with ReLU can.

**Definition 1** (Convex Hull of a Set). *We define the convex hull* $\mathrm{Conv}(P)$ *of $m$ points $P \subset \mathbb{R}^n$ to be* $\mathrm{Conv}(P) = \left\{P\mathbf{a}, \mathbf{a}\in\mathbf{\Delta}^{n-1}\right\}$, *where* $\mathbf{\Delta}^{n-1} = \{\mathbf{a}\in\mathbb{R}^n, a_i \geq 0, \sum_i a_i = 1\}$. *A row $p_j$ is called vertex if $p_j \notin \mathrm{Conv}(P\backslash p_j)$.*

**Definition 2.** *A matrix $P$ of size $m$-by-$n$ is called $k$-vert, or $\mathrm{vert}(P) = k \leq m$, if its $k$ rows are vertices of the convex hull generated by its rows. $P$ is called all-vert if $k = m$.*

**Theorem 6** (Expressibility of ReLU Nonlinearity). *Assuming $m_\alpha = n_\alpha = \mathcal{O}(\exp(\mathrm{sz}(\alpha)))$, where $\mathrm{sz}(\alpha)$ is the size of receptive field of $\alpha$. If each $P_{\alpha\beta}$ is all-vert, then: ($\omega$ is top-level receptive field)*

$$\min_W Loss_{\mathrm{ReLU}}(W) = 0, \quad \min_W Loss_{\mathrm{Linear}}(W) = \mathcal{O}(\exp(\mathrm{sz}(\omega))) \tag{86}$$

*Here we define $Loss(W) \equiv \|F_\omega - I\|_F^2$.*

*Proof.* We prove that in the case of nonlinearity, there exists a weight so that the activation $F_\alpha = I$ for all $\alpha$. We prove by induction. The base case is trivial since we already know that $F_\alpha = I$ for all leaf regions.

Suppose $F_\beta = I$ for any $\beta \in \mathrm{ch}(\alpha)$. Since $P_{\alpha\beta}$ is all-vert, every row is a vertex of the convex hull, which means that for $i$-th row $p_i$, there exists a weight $\mathbf{w}_i$ and $b_i$ so that $\mathbf{w}_i^T p_i + b_i = 1/|\mathrm{ch}(\alpha)| > 0$ and $\mathbf{w}_i^T p_j + b_i < 0$ for $j \neq i$. Put these weights and biases together into $W_{\beta\alpha}$ and we have

$$F_\alpha^{\mathrm{raw}} = \sum_\beta P_{\alpha\beta} F_\beta W_{\beta\alpha} = \sum_\beta P_{\alpha\beta} W_{\beta\alpha} \tag{87}$$

All diagonal elements of $F_\alpha^{\mathrm{raw}}$ are 1 while all off-diagonal elements are negative. Therefore, after ReLU, $F_\alpha = I$. Applying induction, we get $F_\omega = I$ and $G_\omega = I - F_\omega = 0$. Therefore, $Loss_{\mathrm{ReLU}}(W) = \|G_\omega\|_F^2 = 0$.

In the linear case, we know that $\mathrm{rank}(F_\alpha) \leq \sum_\beta \mathrm{rank}(P_{\alpha\beta} F_\beta W_{\beta\alpha}) \leq \sum_\beta \mathrm{rank}(F_\beta)$, which is on the order of the size $\mathrm{sz}(\alpha)$ of $\alpha$'s receptive field (Note that the constant relies on the overlap between

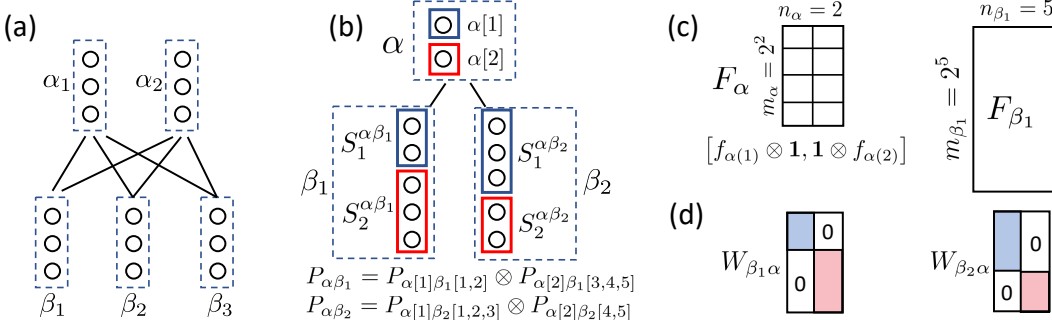

Figure 9: Disentangled representation. **(a)** Nodes are grouped according to regions. **(b)** An example of one parent region $\alpha$ (2 nodes) and two child regions $\beta_1$ and $\beta_2$ (5 nodes each). We assume factorization property of data distribution $P$. **(c)** disentangled activations, **(d)** Separable weights.

receptive fields). However, at the top-level, $m_\omega = n_\omega = \mathcal{O}(\exp(\mathrm{sz}(\omega)))$, i.e., the information contained in $\alpha$ is exponential with respect to the size of the receptive field. By Eckart–Young–Mirsky theorem, we know that there is a lower bound for low-rank approximation. Therefore, the loss for linear network $Loss_\text{linear}$ is at least on the order of $m_0$, i.e., $Loss_\text{linear} = \mathcal{O}(m_\omega)$. Note that this also works if we have BN layer in-between, since BN does a linear transform in the forward pass. $\qquad\square$

This shows the power of nonlinearity, which guarantees full rank of output, even if the matrices involved in the multiplication are low-rank. The following theorem shows that for intermediate layers whose input is not identity, the all-vert property remains.

**Theorem 7.** **(1)** *If $F$ is full row rank, then* $\mathrm{vert}(PF) = \mathrm{vert}(P)$. **(2)** *$PF$ is all-vert iff $P$ is all-vert.*

*Proof.* For (1), note that each row of $PF$ is $p_i^T F$. If $F$ is row full rank, then $F$ has pseudo-inverse $F'$ so that $FF' = I$. Therefore, if $p_i$ is not a vertex:

$$p_i = \sum_{j\neq i} a_j p_j, \quad \sum_j a_j = 1, a_j \geq 0, \tag{88}$$

then $p_i^T F$ is also not a vertex and vice versa. Therefore, $\mathrm{vert}(PF) = \mathrm{vert}(P)$. (2) follows from (1). $\qquad\square$

This means that if all $P_{\alpha\beta}$ are all-vert and its input $F_\beta$ is full-rank, then with the same construction of Thm. 6, $F_\alpha$ can be made identity. In particular, if we sample $W$ randomly, then with probability 1, all $F_\beta$ are full-rank, in particular the top-level input $F_1$. Therefore, using top-level $W_1$ alone would be sufficient to yield zero generalization error, as shown in the previous works that random projection could work well.

### 8.7.2 DISENTANGLED REPRESENTATION

The analysis in the previous section assumes that $n_\alpha = m_\alpha$, which means that we have sufficient nodes, *one neuron for one event*, to convey the information forward to the classification level. In practice, this is never the case. When $n_\alpha \ll m_\alpha = \mathcal{O}(\exp(\mathrm{sz}(\alpha)))$ and the network needs to represent the information in a proper way so that it can be sent to the top level. Ideally, if the factor $z_\alpha$ can be written down as a list of binary factors: $z_\alpha = \left[z_{\alpha[1]}, z_{\alpha[2]}, \ldots, z_{\alpha[j]}\right]$, the output of a node $j$ could represent $z_{\alpha[j]}$, so that all $m_\alpha$ events can be represented concisely with $n_\alpha$ nodes.

To come up with a complete theory for disentangled representation in deep nonlinear network is far from trivial and beyond the scope of this paper. In the following, we make an initial attempt by constructing factorizable $P_{\alpha\beta}$ so that disentangled representation is possible in the forward pass. First we need to formally define what is disentangled representation:

**Definition 3.** *The activation $F_\alpha$ is disentangled, if its $j$-th column $F_{\alpha,:j} = \mathbf{1} \otimes \ldots \otimes \mathbf{f}_{\alpha[j]} \otimes \ldots \otimes \mathbf{1}$, where each $\mathbf{f}_{\alpha[j]}$ and $\mathbf{1}$ is a 2-by-1 vector.*

**Definition 4.** *The gradient $\tilde{G}_\alpha$ is disentangled, if its $j$-th column $\tilde{G}_{\alpha,:j} = \mathbf{p}_{\alpha[1]} \otimes \ldots \otimes \tilde{\mathbf{g}}_{\alpha[j]} \otimes \ldots \otimes$ $\mathbf{p}_{\alpha[n_\alpha]}$, where $\mathbf{p}_{\alpha[j]} = [\mathbb{P}(\alpha[j] = 0), \mathbb{P}(\alpha[j] = 1)]^T$ and $\tilde{\mathbf{g}}_{\alpha[j]}$ is a 2-by-1 vector.*

Intuitively, this means that each node $j$ represents the binary factor $z_\alpha[j]$. A follow-up question is whether such disentangled properties carries over layers in the forward pass. It turns out that the disentangled structure carries if the data distribution and weights have compatible structures:

**Definition 5.** *The weights* $W_{\beta\alpha}$ is separable *with respect to a disjoint set* $\{S_i^{\alpha\beta}\}$, *if* $W_{\beta\alpha} =$ $\mathrm{diag}\left(W_{\beta\alpha}[S_1^{\alpha\beta}, 1], W_{\beta\alpha}[S_2^{\alpha\beta}, 2], \ldots, W_{\beta\alpha}[S_{n_\alpha}^{\alpha\beta}, n_\alpha]\right)$.

If the bottom activations are disentangled, by induction, all activations should be disentangled. The next question is whether gradient descent preserves such a structure. Here we provide a few theorems to discuss such issues.

We first start with two lemmas. Both of them have simple proofs.

**Lemma 1.** *Distribution representations have the following property:*

(1) *If* $F_\alpha^{(i)}$ *is disentangled,* $F_\alpha = \sum_i w_i F_\alpha^{(i)}$ *is also disentangled.*

(2) *If* $F_\alpha$ *is disentangled and* $h$ *is any per-column element-wise function, then* $h(F_\alpha)$ *is disentangled.*

(3) *If* $F_\alpha^{(i)}$ *are disentangled,* $h_i$ *are per-column element-wise function, then* $h_1(F_\alpha^{(1)}) \circ h_2(F_\alpha^{(2)}) \ldots \circ h_n(F_\alpha^{(n)})$ *is disentangled.*

*Proof.* (1) follows from properties of tensor product. For (2) and (3), note that the $j$-th column of $F_\alpha$ is $F_{\alpha,:j} = \mathbf{1} \otimes \ldots \mathbf{f}_j \ldots \otimes \mathbf{1}$, therefore $h^j(F_{\alpha,:j}) = \mathbf{1} \otimes \ldots h^j(\mathbf{f}_j) \ldots \otimes \mathbf{1}$, and $h_1^j(F_{\alpha,:j}^{(1)}) \circ h_2^j(F_{\alpha,:j}^{(2)}) = \mathbf{1} \otimes \ldots h_1^j(\mathbf{f}_j^{(1)}) \circ h_2^j(\mathbf{f}_j^{(2)}) \ldots \otimes \mathbf{1}$. $\square$

Given one child $\beta \in \mathrm{ch}(\alpha)$, denote

$$P_{S_j} = P_{\alpha[j]\beta[S_j]} = \left[\mathbb{P}(z_{\beta[S_j]}|z_{\alpha[j]})\right] \tag{89}$$

$$\mathbf{w}_{S_j} = W_{\beta\alpha}[S_j, j] \tag{90}$$

$$\mathbf{p}_{\alpha[j]} = \left[\mathbb{P}(\alpha[j]=0), \mathbb{P}(\alpha[j]=1)\right]^T \tag{91}$$

We have $P_{S_j}\mathbf{1} = \mathbf{1}$ and $\mathbf{1}^T\mathbf{p}_{\alpha[j]} = 1$. Note here for simplicity, $\mathbf{1}$ represents all-one vectors of any length, determined by the context.

Since $F_\alpha$ and $G_\beta$ are disentangled, their $j$-th column can be written as:

$$F_{\beta,:j} = \mathbf{1} \otimes \ldots \otimes \mathbf{f}_j \otimes \ldots \otimes \mathbf{1} \tag{92}$$

$$\tilde{G}_{\alpha,:j} = \mathbf{p}_{\alpha[1]} \otimes \ldots \otimes \tilde{\mathbf{g}}_j \otimes \ldots \otimes \mathbf{p}_{\alpha[n_\alpha]} \tag{93}$$

For simplicity, in the following proofs, we just show the case that $n_\alpha = 2, n_\beta = 3, z_\alpha = \left[z_{\alpha[1]}, z_{\alpha[2]}\right]$ and $S = \{S_1, S_2\} = \{\{1, 2\}, \{3\}\}$. We write $\mathbf{f}_{1,2} = [\mathbf{f}_1 \otimes \mathbf{1}, \mathbf{1} \otimes \mathbf{f}_2]$ as a 2-column matrix. The general case is similar and we omit here for brevity.

**Theorem 8** (Disentangled Forward). *If for each* $\beta \in \mathrm{ch}(\alpha)$, $P_{\alpha\beta}$ *can be written as a tensor product* $P_{\alpha\beta} = \bigotimes_i P_{\alpha[i]\beta[S_i^{\alpha\beta}]}$ *where* $\{S_i^{\alpha\beta}\}$ *are* $\alpha\beta$-*dependent disjointed set,* $W_{\beta\alpha}$ *is separable with respect to* $\{S_i^{\alpha\beta}\}$, $F_\beta$ *is disentangled, then* $F_\alpha$ *is also disentangled (with/without ReLU /Batch Norm).*

*Proof.* For a certain $\beta \in \mathrm{ch}(\alpha)$, we first compute the quantity $P_{\alpha\beta}F_\beta$:

$$P_{\alpha\beta}F_\beta = (P_{1,2} \otimes P_3)\left[\mathbf{f}_{1,2} \otimes \mathbf{1}, \quad \mathbf{1} \otimes \mathbf{f}_3\right] = [P_{1,2}\mathbf{f}_{1,2} \otimes \mathbf{1}, \quad \mathbf{1} \otimes P_3\mathbf{f}_3] \tag{94}$$

Therefore, the forward information sent from $\beta$ to $\alpha$ is:

$$F_{\beta\to\alpha}^{\mathrm{raw}} = P_{\alpha\beta}F_\beta W_{\beta\alpha} = [P_{1,2}\mathbf{f}_{1,2} \otimes \mathbf{1}, \quad \mathbf{1} \otimes P_3\mathbf{f}_3]\begin{bmatrix} \mathbf{w}_{1,2} & 0 \\ 0 & \mathbf{w}_3 \end{bmatrix} \tag{95}$$

$$= [P_{1,2}\mathbf{f}_{1,2}\mathbf{w}_{1,2} \otimes \mathbf{1}, \quad \mathbf{1} \otimes P_3\mathbf{f}_3\mathbf{w}_3] \tag{96}$$

Note that both $P_{1,2}\mathbf{f}_{1,2}\mathbf{w}_{1,2}$ and $P_3\mathbf{f}_3\mathbf{w}_3$ are 2-by-1 vectors. Therefore, for each $\beta \in \mathrm{ch}(\alpha)$, $F_{\beta \to \alpha}^{\mathrm{raw}}$ is disentangled. By Lemma 1, both $F_\alpha^{\mathrm{raw}} = \sum_{\beta \in \mathrm{ch}(\alpha)} F_{\beta \to \alpha}^{\mathrm{raw}}$ and the nonlinear response $F_\alpha$ are disentangled. By Eqn. 83, the forward pass of Batch Norm is a per-column element-wise function, so BN also preserves disentangledness. $\qquad\square$

**Theorem 9** (Separable Weight Update). *If $P_{\alpha\beta} = \bigotimes_i P_{\alpha[i]\beta[S_i]}$, both $F_\beta$ and $\tilde{G}_\alpha$ are disentangled, $\mathbf{1}^T \tilde{G}_\alpha = \mathbf{0}$, then the gradient update $\Delta W_{\beta\alpha}$ is separable with respect to $\{S_i\}$.*

*Proof.* Following Eqn. 62 and Eqn. 94, we have:

$$\Delta W_{\beta\alpha} = (P_{\alpha\beta}F_\beta)^T \tilde{G}_\alpha \tag{97}$$

$$= \left[ \begin{array}{c} (P_{1,2}\mathbf{f}_{1,2})^T \otimes \mathbf{1}^T \\ \mathbf{1}^T \otimes (P_3\mathbf{f}_3)^T \end{array} \right] \left[ \tilde{\mathbf{g}}_1 \otimes \mathbf{p}_{\alpha[2]}, \quad \mathbf{p}_{\alpha[1]} \otimes \tilde{\mathbf{g}}_2 \right] \tag{98}$$

$$= \left[ \begin{array}{cc} (P_{1,2}\mathbf{f}_{1,2})^T \tilde{\mathbf{g}}_1 & (P_{1,2}\mathbf{f}_{1,2})^T \mathbf{p}_{\alpha[1]} \otimes \mathbf{1}^T \tilde{\mathbf{g}}_2 \\ \mathbf{1}^T \tilde{\mathbf{g}}_1 \otimes (P_3\mathbf{f}_3)^T \mathbf{p}_{\alpha[2]} & (P_3\mathbf{f}_3)^T \tilde{\mathbf{g}}_2 \end{array} \right] \tag{99}$$

Since $\mathbf{1}^T \tilde{G}_\alpha = \mathbf{0}$, we have for any $j$, $\mathbf{1}^T \tilde{G}_{\alpha,:j} = 0$ and thus $\mathbf{1}^T \tilde{\mathbf{g}}_j = 0$. Therefore,

$$\Delta W_{\beta\alpha} = \mathrm{diag}\left( (P_{1,2}\mathbf{f}_{1,2})^T \tilde{\mathbf{g}}_1, \quad (P_3\mathbf{f}_3)^T \tilde{\mathbf{g}}_2 \right) \tag{100}$$

which is separable with respect to $S$. In particular:

$$\Delta \mathbf{w}_{1,2} = (P_{1,2}\mathbf{f}_{1,2})^T \tilde{\mathbf{g}}_1, \quad \Delta \mathbf{w}_3 = (P_3\mathbf{f}_3)^T \tilde{\mathbf{g}}_2 \tag{101}$$

$\qquad\square$

### 8.7.3 DISCUSSION ABOUT BACKPROPAGATION OF DISENTANGLED GRADIENT

One problem remains. If $\{\tilde{G}_\alpha\}_{\alpha \in \mathrm{pa}(\beta)}$ are all disentangled, whether $\tilde{G}_\beta$ is disentangled? We can try computing the following quality:

$$\tilde{G}_{\alpha \to \beta}^{\mathrm{raw}} = P_{\alpha\beta}^T \tilde{G}_\alpha W_{\beta\alpha}^T \tag{102}$$

$$= \left( P_{1,2}^T \otimes P_3^T \right) \left[ \tilde{\mathbf{g}}_1 \otimes \mathbf{p}_{\alpha[2]}, \quad \mathbf{p}_{\alpha[1]} \otimes \tilde{\mathbf{g}}_2 \right] \left[ \begin{array}{cc} \mathbf{w}_{1,2}^T & 0 \\ 0 & \mathbf{w}_3^T \end{array} \right] \tag{103}$$

$$= \left[ P_{1,2}^T \tilde{\mathbf{g}}_1 \otimes \mathbf{p}_{\beta[3]}, \quad \mathbf{p}_{\beta[1,2]} \otimes P_3^T \tilde{\mathbf{g}}_2 \right] \left[ \begin{array}{cc} \mathbf{w}_{1,2}^T & 0 \\ 0 & \mathbf{w}_3^T \end{array} \right] \tag{104}$$

$$= \left[ P_{1,2}^T \tilde{\mathbf{g}}_1 \mathbf{w}_{1,2}^T \otimes \mathbf{p}_{\beta[3]}, \quad \mathbf{p}_{\beta[1,2]} \otimes P_3^T \tilde{\mathbf{g}}_2 \mathbf{w}_3^T \right] \tag{105}$$

Note that here we use the following equality from total probability rule:

$$P_3^T \mathbf{p}_{\alpha[2]} = \mathbf{p}_{\beta[3]}, \quad P_{1,2}^T \mathbf{p}_{\alpha[1]} = \mathbf{p}_{\beta[1,2]} \tag{106}$$

where $\mathbf{p}_{\beta[1,2]}$ is a 4-by-1 vector:

$$\mathbf{p}_{\beta[1,2]} = \left[ \begin{array}{c} \mathbb{P}(z_{\beta[1]} = 0, z_{\beta[2]} = 0) \\ \mathbb{P}(z_{\beta[1]} = 0, z_{\beta[2]} = 1) \\ \mathbb{P}(z_{\beta[1]} = 1, z_{\beta[2]} = 0) \\ \mathbb{P}(z_{\beta[1]} = 1, z_{\beta[2]} = 1) \end{array} \right] \tag{107}$$

Note that the ordering of these joint probability corresponds to the column order of $P_{1,2}$.

Now with this example, we see that the backward case ( Eqn. 106) is very different from the forward case (Eqn. 96), in which $\tilde{G}_{\alpha \to \beta}^{\mathrm{raw}}$ is no longer disentangled. Indeed, $P_{1,2}^T \tilde{\mathbf{g}}_1 \mathbf{w}_{1,2}^T$ is a 2-column matrix and $\mathbf{p}_{\beta[1,2]}$ is not a rank-1 tensor anymore. Intuitively this makes sense, if two low-level attributes have very similar behaviors, there is no way to distinguish the two via backpropagation.

Note that we also *cannot* assume independence: $\mathbf{p}_{\beta[1,2]} = \mathbf{p}_{\beta[1]} \otimes \mathbf{p}_{\beta[2]}$ since the independence property is in general not carried from layer to layer.

For general cases, $\tilde{G}_{\alpha\to\beta}^{\mathrm{raw}}$ takes the following form:

$$\tilde{G}_{\alpha\to\beta}^{\mathrm{raw}} = \left[ P_{S_1^{\alpha\beta}}^T \tilde{\mathbf{g}}_1^\alpha \mathbf{w}_{S_1^{\alpha\beta}}^T \otimes \bigotimes_{j=2}^{n_\alpha} \mathbf{P}_{\beta[S_j^{\alpha\beta}]}, \quad \mathbf{P}_{\beta[S_1^{\alpha\beta}]} \otimes P_{S_2^{\alpha\beta}}^T \tilde{\mathbf{g}}_2^\alpha \mathbf{w}_{S_2^{\alpha\beta}}^T \otimes \bigotimes_{j=3}^{n_\alpha} \mathbf{P}_{\beta[S_j^{\alpha\beta}]}, \quad \cdots \right]$$

(108)

One hope here is that if we consider $\sum_{\alpha\in\mathrm{pa}(\beta)} \tilde{G}_{\alpha\to\beta}^{\mathrm{raw}}$, the summation over parent $\alpha$ could lead to a better structure, even for individual $\alpha$, $P_{S_1}^T \tilde{\mathbf{g}}_1^\alpha \mathbf{w}_{S_1}^T$ is not 1-order tensor. For example, if $S_j^{\alpha\beta} = S_j$, then for the first column in $S_1$, due to $\mathbf{1}^T \tilde{\mathbf{g}}_j = 0$, we know that:

$$\sum_{\alpha\in\mathrm{pa}(\beta)} P_{\alpha,S_1}^T \tilde{\mathbf{g}}_1^\alpha \mathbf{w}_{\alpha,S_1[1]}^T = \sum_{\alpha\in\mathrm{pa}(\beta)} c_\alpha (\mathbf{v}_{\alpha,S_1}^+ - \mathbf{v}_{\alpha,S_1}^-)$$

(109)

where $\mathbf{v}_{\alpha,S_1}^+ = \mathbb{P}(z_{\beta[S_1]} | z_{\alpha[1]} = 1)$ and $\mathbf{v}_{\alpha,S_1}^- = \mathbb{P}(z_{\beta[S_1]} | z_{\alpha[1]} = 0)$ and $P_{\alpha,S_i} = \begin{bmatrix} \mathbf{v}_{\alpha,S_1}^- \\ \mathbf{v}_{\alpha,S_1}^+ \end{bmatrix}$.

If each $\alpha \in \mathrm{pa}(\beta)$ is informative in a diverse way, and $|S_1|$ is relatively small (e.g., 4), then $\mathbf{v}_{\alpha,S_1}^+ - \mathbf{v}_{\alpha,S_1}^- \neq \mathbf{0}$ and spans the probability space of dimension $2^{|S_1|} - 1$. Then we can always find $c_\alpha$ (or equivalently, weights) so that Eqn. 109 becomes rank-1 tensor (or disentangled). Besides, the gating $D_\beta$, which is disentangled as it is an element-wise function of $F_\beta$, will also play a role in regularizing $\tilde{G}_\beta$.

We will leave this part to future work.

