# OpenReview forum: "A theoretical framework for deep and locally connected ReLU network"
_ICLR.cc/2019/Conference_

### Official Review · AnonReviewer2 · 2018-11-02

**Rating:** 5
**Confidence:** 3

**Review:**

This paper gives a model for understanding locally connected neural networks. The main idea seems to be that the network is sparsely connected, so each neuron is not going to have access to the entire input. One can then think about the gradient of this neuron locally while average out over all the randomness in the input locations that are not relevant to this neuron. Using this framework the paper tried to explain several phenomena in neural networks, including batch normalization, overfitting, disentangling, etc.

I feel the paper is poorly written which made it very hard to understand. For example, as the paper states, the model gives a generative model for input (x,y) pairs. However, I could not find a self-contained description of how this generative model works. Some things are described in Section 3.1 about the discrete summarization variables, but the short paragraph did not describe: (a) What is the "multi-layer" deterministic function? (b) How are these z_\alpha's chosen? (c) Given z's how do we generate x? (d) What happens if we have z_\alpha and z_\beta and the regions \alpha and \beta are not disjoint? What x do we use in the intersection?

In trying to understand the paper, I was thinking that (a)(b) The multilayer deterministic function is a function which gives a tree structure over the z_\alpha's, where y is the root. (I have no idea why this should be a deterministic function, intuitively shouldn't y be chosen randomly, and each z_\alpha chosen randomly conditioned on its parent?)  (c) there is a fixed conditional distribution of P(x_\alpha|z_\alpha), and I really could not figure out (d). The paper definitely seems to allow two receptive fields to intersect as in Figure 1(b).

Without understanding the generative model, it is impossible for me to evaluate the later results. My general comments there is that there are no clear Theorems that summarizes the results (the Theorems in the paper are all just Lemmas that are trying to work towards the final goal of giving some explanations, but the explanations and assumptions are not formally written down). Looking at things separately (as again I couldn't understand the single paragraph describing the generative model), the Assumption in Theorem 3 seems extremely limiting as it is saying that x_j is a discrete distribution (which is probably never true in practice). I wouldn't say "the model does not impose unrealistic assumptions" in abstract if you are going to assume this, rather the model just makes a different kind of unrealistic assumptions (Assumptions in Theorem 2 might be much weaker, but it's hard to judge that).

==== After reading the revision

The revised version is indeed more clear about how the teacher network works, and I have tried to understand the later parts of the paper again. The result of the paper really relies on the two assumptions in Theorem 2. Of the two assumptions, the first one seems to be intuitive (and it is OK although exact conditional independence might be slightly strong). The second assumption is very unclear though as it is not an assumption that is purely about the model/teacher network (which are the x and z variables), it also has to do with the learning algorithm/student network (f's and g's). It is much harder to reason about the behavior of an algorithm on a particular model and directly making an assumption about that in some sense hides the problem. The paper mentioned that the condition is true if z is fine-grained, but this is very vague - it is definitely true if z is super fine-grained to satisfy the assumption in Theorem 3, but that is too extreme.

Overall I still feel the paper is a bit confusing and it would benefit from having a more concrete example. I like the direction of the work but I can't recommend for recommendation at this stage.

---

### Official Review · AnonReviewer1 · 2018-11-02
**The authors propose a framework that utilizes the teacher-student setting and give some impressive evaluations on deep neural networks. This paper has rigorous theoretical analysis, but lacks necessary experiments.**

**Rating:** 7
**Confidence:** 4

**Review:**

The authors propose a framework that utilizes the teacher-student setting to evaluate deep locally connected ReLU network. The framework explicitly formulates data distribution, which has not been considered by previous works. The authors also show that their framework is compatible with Batch Normalization and favors disentangled representation when data distributions have factorizable structures. Based on this framework, the authors re-explain some common issues of deep learning, such as overfitting.

My major concerns are as follows.

1. The framework is based on the teacher-student setting, and the authors claim that "the teacher generates classification label via a hidden computational graph". However, how the teacher can be designed is not clear in the paper.

2. The data distribution included in this paper is $P(z_{\alpha}, z_{\beta})$, where $z_{\alpha}$ and $z_{\beta}$ are all summarization variables. From this perspective, it only has an indirect connection with original data distribution $P(x)$ or $P(x_{\alpha}, x_{\beta})$, and thus it could be questionable whether $P(z_{\alpha}, z_{\beta})$ is a convincing representation.

3. The authors may want to conduct more experiments to better support their claims.

---

### Official Review · AnonReviewer3 · 2018-11-04
**This paper proposes a new framework for understanding the Relu networks in theory. However, the assumptions are not justified and the definitions seems not clear.**

**Rating:** 3
**Confidence:** 4

**Review:**

This paper proposes a new approach to understand the theory of RELU neural networks. Using a teacher-student setting, this paper studies the batch normalization and the disentangled representations of neural networks. However, the definitions of some of the concepts and notation are not sufficiently clear. In addition, the assumptions that the main results of this paper depend on do not have clear intuitions.

Detailed comments:

1. It seems that this paper over claims its contribution. It is not clear why the "teacher-student setting" can be called a theoretical framework, even the definitions of the teacher and the student are not clear. It seems that the new framework is just a way to compute the relations of the gradients of neurons based on a few assumptions (Theorem 2).

2. I found it very hard to follow the notations given in this paper. The main reason is that many of the terms appear without a definition, and the reader has to guess what they stand for. For example, in equation (2), w_{jk} seems to be the weight between nodes j and k, where k is a child of j. But this term is not defined. As another example, all the matrices in Theorem 9 are not defined. They just suddenly appear. In addition, S(f) in (11) is not defined. I would suggest the authors to spend one section to carefully define everything.

3. The theorems all depends on some assumptions that are unclear whether will hold in practice or not. For example, in theorem 2, it is hard to see what kind of data distribution satisfy these three conditions. Although in Theorem 3 the author gave a sufficient condition, we still don't know what kind of $X$ satisfies this. For example, does Gaussian distribution satisfy those? This problem also happens to other theorems. It would be much better to make sure that these assumptions are unrealistic.

---

### Public Comment · (anonymous) · 2018-10-01
**Insufficient Exposition of Previous Works**

This is really an interesting and technical theoretical paper. I've added this submission on my reading list and detailed comments will come later. However, to my knowledge, the references in this paper are very insufficient and some related works are not cited. To my knowledge, the author should cite papers, for example, like the following,

Achille, Alessandro, and Stefano Soatto. "Emergence of invariance and disentangling in deep representations."

Another thing I'm concerned is that if this is not a seminar paper to our deep learning theory community, the title should not contain words like "framework". My question is, "Have you really proposed a mathematically rigorous framework for deep ReLU network? "

---

> ### Author Response · Authors · 2018-10-01
> **Will add more related works in the next revision.**
>
> Thanks for your comment!
>
> We appreciate your interest to our paper.
>
> We totally agree that there should be more related works in the submission, in particular for the great work on theoretical foundation of information bottleneck. We will add them in the next revision.
>
> We want to emphasize that there is one substantial difference between our work and the works of information bottleneck: we model data distribution as explicit terms in our reformulation of deep and locally connected nonlinear network. To our best knowledge, this is novel.  By imposing different conditions on the data distribution, there could be many interesting consequences. In our paper, we only barely scratch the surface.
>
> For mathematical rigorousness: Given the assumptions in the paper are true, to our best knowledge and efforts, all the statements named "theorems" in our paper are rigorous. You can check the Appendix for all the detailed proofs. Note that it is totally possible that we might make mistakes. If so we would happily revise the paper and/or retract.
>
> We are looking forward to your detailed comments.

---

### Author Response · Authors · 2018-11-08
**Rebuttal**

We thank the reviewers for their insightful comments.

We acknowledge that there is confusion in terms of paper organization and math notations. We are working on a revision, which will be uploaded by Nov. 23.

For now we first address the main questions raised by the reviewers.

1. [R2] There is no main theorem in this paper.

It remains a grand challenge for the whole community to come up with a theorem that relates data distribution to the properties of deep and nonlinear network. As the major contribution and a first step, we propose a reformulation that explicitly relates data distribution to the gradient descent optimization procedure. With this reformulation, we now can explicitly study how data distribution affects the property of the network. Along this direction, we put a few initial discussions in the paper. An application of this reformulation towards a major theorem is left to future work.

Besides, we also discover a property of back-propagated gradient of BatchNorm (Sec. 4) and show that this property is preserved in the reformulation.

2. [R1][R2] How teacher is defined.

The teacher is specified in a bottom-up manner (rather than top-down, as suggested by the reviewer 2). First the lowest layers of summarization is computed, then the second lowest layer of summarization is computed based on the lowest layers (ref. Sec 3.1: z_\alpha = z_\alpha(z_\beta)), until the top-level summarization is computed, which is the class label y. At each stage, we assume that the upwards function be deterministic and typically drop irrelevant information w.r.t the class label. The reason why we want a deterministic function is for the proof of Theorem 2. Note that while top-down graphical model requires nondeterministic function (since new information needs to be added), assuming deterministic function in a bottom-up setting is natural.

3. [R2] How z_\alpha is picked and how to define P(x_\alpha|z_\alpha):

z_\alpha is picked by the teacher. Since the teacher provides the classification labels for the student, any choice of z_\alpha would fit to the theory. Intuitively, z_\alpha is a summarization of the content x_\alpha within the receptive field \alpha, which contributes to the final label y.

Due to a loss of information, there are multiple x_\alpha that maps to the same z_\alpha. Therefore, we can define P(x_\alpha|z_\alpha).

4. [R2][R3] Is the assumption in Theorem 2 (and 3) realistic?

Typically, z_\alpha and z_\beta are overlapping (Fig. 1(b)). In particular, if \alpha is a parent of \beta, then the receptive field \alpha covers \beta. With this in mind, the two assumptions in Theorem 2:

          P(x_\alpha|z_\alpha, z_\beta) = P(x_\alpha|z_\alpha)

                 and

          P(x_\beta|z_\alpha, z_\beta) = P(x_\beta|z_\beta)

are natural, since each z is most related to the information of its own receptive field.

Theorem 3 is a limiting case of Theorem 2, which gives an example about when the assumptions of theorem 2 hold exactly. If the assumptions of Theorem 2 are relaxed (e.g., ||P(x_\alpha|z_\alpha, z_\beta) - P(x_\alpha|z_\alpha)|| \le \epsilon), still we have bounds (instead of equalities) in Theorem 2.

5. [R1] The data distribution is indirectly characterized by the conditional distribution P(z_\alpha | z_\beta), which may not be ideal/questionable.

We think it is more like a merit rather than a shortcoming. An indirect specification (like what we give in the paper) gives much more flexibility of the distribution x. In contrast, a direct/parametric specification (e.g., the input data is Gaussian) might look mathematically clear but is probably not true in practice. On the other hand, given this indirect specification, we agree that the resulting distribution of input deserves further empirical study (e.g., via sampling x given z).

 6. [R1] Empirical study
We will add more empirical studies in the next revision. We already observe the convergence of the reformulations (Theorem 2) under random conditional distribution of summarization variable (P(z_\alpha|z_\beta)), and much faster convergence if BatchNorm is used.

---

> ### Comment · AnonReviewer2 · 2018-11-08
> **thanks for the clarification**
>
> After the clarification the model seems to be making more sense (on the other hand I really couldn't find how the teacher is defined in the original version, looking forward to your updated version). I will try to evaluate the paper again once the revision is uploaded.

---

### Author Response · Authors · 2018-11-20
**Revision (v2)**

We have updated the main text of our paper to make the writing more clear. Please take a look.

1. Our teacher-student setting is introduced with more explanation and examples (e.g., a conceptual comparison between our modeling and top-down generative model).

2. We have added more related works.

3. The assumptions in Theorem 2 are explained, showing they are mild assumptions.

4. The relationship between input data distribution P(x) and the conditional distribution of P(z_\beta|z_\alpha) is explained.

5. We put a table (Tbl. 2) explaining all notations used in Sec. 5.2.

6. A better explanation of how backpropagation of BatchNorm was regarded as a projection and how it is compatible with our framework. S(f) now is defined.

Due to time limit, we will add empirical study in the camera ready (if there is one).

---

> ### Comment · AnonReviewer3 · 2018-11-29
> **The notation is still not clearly explained.**
>
> I appreciate the effort of the authors to make the paper more accessible. However, in the updated version, it seems that some notation is still not clearly defined or explained. Some examples are as follows. In equation (2), the notation $g_j(x_k)$ seems to coincide with $g_j(x)$, which might be misleading. In Theorem 1, what is the relationship between j and k? It would be better to elaborate more on how to use the recursive relations to compute the marginal gradient.
> Moreover, on page 4, Equ.20 should be Equ. 4.
>
> My main concern about Theorem 2 is that the assumption might not hold. Still, in the revised version, the authors are unable to come up with an example that shows these abstract assumptions are true.
>
> My another concern is with the assumption of the locally connected neural network. It seems that the whole derivation hinges on this assumption.
>
> Furthermore, the pivotal part of the framework is the Marginalized Gradient, which is defined by taking conditional expectations of the input of each layer. However, the weights of the neural network depend on the input data, which makes it unable to take expectation as if the weights are deterministic. Treating these weights as deterministic numbers are only possible if you consider the SGD setting where you use a fresh sample to evaluate the expectation. Even in this case, the expectation should be taken only with respect to the input data. However, it seems that, in the paper, expectations are taken with respect to the input and outputs of each neuron.

---

> > ### Author Response · Authors · 2018-11-30
> > **Clear some confusions**
> >
> > Thanks R3 for taking time to read the revised paper! We really appreciated it.
> >
> > The assumption of "locally connected neural network" is indeed important. However we regard this assumption as an important contribution of this paper rather than a restriction. First of all, network with such structures (e.g., CNN) indeed works in practice but how they work remain an open problem. Therefore, a theoretical analysis is important. Furthermore, without such structural assumptions, a general analysis of neural networks won't lead to meaningful conclusion that connects to what we see in practice. Our paper is an attempt to make such connections.
> >
> > The assumptions in Theorem 2 are intuitive and are explained right after Theorem 2. For the first assumption, the intuition is: the summarization variable z_\alpha at the receptive field \alpha mainly captures the input content x_\alpha at that receptive field. Given that summarization variable z_\alpha, others have minor effects. Therefore, we assume that P(x_\alpha | z_\alpha, z_{others}) = P(x_\alpha | z_\alpha). The second assumption is technical and is true when z_\alpha splits x_\alpha well. In reality, we can always turn the two assumptions into bounds and Theorem 2 becomes bounds as well.
> >
> > We argue that these assumptions are way more natural than parametric forms (e.g., Gaussian inputs), and serve as one step closer to the real situations.
> >
> > Marginalized Gradient: At iteration t, we can always treat the current weights as constants and take expectation over the data distribution. In fact, it is a standard practice in many previous theoretical papers [1-4], no matter whether population gradient or stochastic gradient are used.
> >
> > We didn't take expectation with respect to the output of the neuron. Do you mean the label $y$? In this case, it is not an issue since y is assumed to be the deterministic function of the input x. Therefore, taking expectation with respect to (x, y) is the same as with respect to x (the input) alone. We already explained it in the paper (Page 4, just before "Marginalized Gradient").
> >
> > Finally, the notations $g_j(x_k)$, $g_j(x_j)$ and $g_j(x)$ were intentionally overloaded to make the notation cleaner. Otherwise one would need to use different functions for different version of the gradient signal. Alternatively we could use $g_{j\rightarrow k}(x_k)$, which is a bit heavy in notation (e.g., too many "k"). Therefore, this is more or less a subjective thing.
> >
> > Throughout the paper, it is a convention that node j is the parent of node k. We will mention clearly the role of j and k in Theorem 2 in the next revision.
> >
> > References:
> > [1] R. Ge et al. Learning One-hidden-layer Neural Networks with Landscape Design. https://arxiv.org/abs/1711.00501
> > [2] S. Du et al. Gradient Descent Learns One-hidden-layer CNN: Don't be Afraid of Spurious Local Minima
> > https://arxiv.org/abs/1712.00779
> > [3] A. Brutzkus and A. Globerson, Globally Optimal Gradient Descent for a ConvNet with Gaussian Inputs
> > https://arxiv.org/abs/1702.07966
> > [4] Z. Pan and J. Feng, Empirical Risk Landscape Analysis for Understanding Deep Neural Networks. https://openreview.net/pdf?id=B1QgVti6Z

---

### Meta-Review · Area_Chair1 · 2018-12-17
**Further iteration needed**

**Confidence:** 5
**Recommendation:** Reject

**Metareview:**

This paper studies the behavior of gradient descent on deep neural network architectures with spatial locality, under generic input data distributions, using a planted or "teacher-student" model.

Whereas R1 was supportive of this work, R2 and R3 could not verify the main statements and the proofs due to a severe lack of clarity and mathematical rigor. The AC strongly aligns with the latter, and therefore recommends rejection at this time, encouraging the authors to address clarity and rigor issues and resubmit their work again.